# The SGLT2 inhibitor canagliflozin suppresses growth and enhances prostate cancer response to radiotherapy

Amr Ali [1,2,3], Bassem Mekhaeil[1], Olga-Demetra Biziotis [1,2,3], Evangelia E. Tsakiridis[2,4], Elham Ahmadi [1,2], Jianhan Wu[2,4], Simon Wang[1,2,3], Kanwaldeep Singh [1,3], Gabe Menjolian[5], Thomas Farrell[6], Aruz Mesci[1,7], Stanley Liu [8], Tobias Berg [1,3], Jonathan L. Bramson [1,3,9], Gregory R. Steinberg [2,4,10] & Theodoros Tsakiridis [1,2,3,7,9 ✉]

Radiotherapy is a non-invasive standard treatment for prostate cancer (PC). However, PC develops radio-resistance, highlighting a need for agents to improve radiotherapy response. Canagliflozin, an inhibitor of sodium-glucose co-transporter-2, is approved for use in diabetes and heart failure, but is also shown to inhibit PC growth. However, whether canagliflozin can improve radiotherapy response in PC remains unknown. Here, we show that well-tolerated doses of canagliflozin suppress proliferation and survival of androgen-sensitive and insensitive human PC cells and tumors and sensitize them to radiotherapy. Canagliflozin blocks mitochondrial respiration, promotes AMPK activity, inhibits the MAPK and mTOR-p70$^{S6k}$/4EBP1 pathways, activates cell cycle checkpoints, and inhibits proliferation in part through HIF-1$\alpha$ suppression. Canagliflozin mediates transcriptional reprogramming of several metabolic and survival pathways known to be regulated by ETS and E2F family transcription factors. Genes downregulated by canagliflozin are associated with poor PC prognosis. This study lays the groundwork for clinical investigation of canagliflozin in PC prevention and treatment in combination with radiotherapy.

[1] Departments of Oncology, McMaster University, Hamilton, ON, Canada. [2] Centre for Metabolism, Obesity and Diabetes Research, McMaster University, Hamilton, ON, Canada. [3] Centre for Discovery in Cancer Research, McMaster University, Hamilton, ON, Canada. [4] Departments of Medicine, McMaster University, Hamilton, ON, Canada. [5] Department of Radiotherapy, Juravinski Cancer Center, Hamilton, ON, Canada. [6] Department of Physics, Juravinski Cancer Center, Hamilton, Ontario, Canada. [7] Department of Radiation Oncology, Juravinski Cancer Center, Hamilton, ON, Canada. [8] Odette Cancer Centre, Sunnybrook Health Sciences Centre, University of Toronto, Toronto, ON, Canada. [9] Department of Pathology and Molecular Medicine, McMaster University, Hamilton, ON, Canada. [10] Department of Biochemistry and Biomedical Sciences, McMaster University, Hamilton, ON, Canada. ✉email: tsakirt@mcmaster.ca

Prostate cancer (PC) is the fourth most diagnosed cancer worldwide[1]. There is a need for PC prevention and strategies to improve standard therapy outcomes in this disease. Today, most patients with PC are treated with radiotherapy with or without androgen deprivation therapy[2]. However, a large proportion of PC tumors develops resistance to radiotherapy resulting in disease recurrence, metastasis, morbidity, and mortality[3]. In the past two decades, clinical trials focused on improving outcomes by increasing the dose of radiotherapy delivered to the prostate[4]. However, radiotherapy dose-escalation is associated with increased short- and long-term bowel and genitourinary toxicity (5–20%, RTOG grade 2)[5,6]. The development of agents that can synergize with or sensitize PC to radiotherapy would be highly beneficial.

Work in recent decades illustrated the vital role of metabolic deregulation in tumor cell survival and resistance to cytotoxic therapy[7,8]. In normal prostatic tissue, androgen receptor (AR) signaling guides the utilization of glucose and amino acids through glycolysis and the tricarboxylic acid (TCA) cycle to generate and release citrate in the lumen of prostatic glands[9]. Unlike Warburg's model[10], which suggested a diminishing role of tumor cell mitochondria, PC cells demonstrate enhanced mitochondria function and use the TCA cycle to convert substrate supply from glycolysis, amino acid influx and protein catabolism to de novo synthesis of nucleotides, proteins, and lipids required for cellular growth[11]. The phosphatidylinositol 3-kinase (PI3k)—Akt—mammalian target of rapamycin (mTOR) pathway provides vital support for this function through regulation of membrane transporters, glycolytic and lipogenic enzyme gene expression, while it regulates protein synthesis through p70-S6 kinase (p70$^{S6k}$) and eukaryotic translation initiation factor 4E-binding-protein-1 (4EBP1)[12]. HIF-1$\alpha$ an established cellular response to hypoxia, supports cell survival in the hypoxic tumor microenvironment, but also operates during normoxia downstream of mTOR[13] to promote angiogenesis, radio-resistance, and metastasis[14]. mTOR facilitates HIF-1$\alpha$ expression through 4EBP1 and STAT3[12], but also stabilizes HIF-1$\alpha$ through p70$^{S6k}$-dependent inhibition of the phosphatase PP2A. The latter deactivates the HIF-1$\alpha$ prolyl hydroxylase 2 (PHD2), an enzyme that mediates HIF-1$\alpha$ hydroxylation leading to E3-ubiquitination and degradation[15]. HIF-1$\alpha$ is overexpressed in more than 70% of human cancers and is associated with poor prognosis[16]. Conversely, HIF-1$\alpha$ loss inhibits tumor growth in xenograft studies[17].

On this basis, targeting mitochondrial metabolism is an attractive strategy to curtail PC growth. This strategy, however, has additional merit. Mitochondrial inhibition leads to the activation of the metabolic stress sensor AMP-activated kinase (AMPK), a hetero-trimeric enzyme with alpha-catalytic and beta- and gamma-regulatory subunits[18]. AMPK is activated through AMP/ADP binding to its gamma-subunit leading to inhibition of, (i) mTOR complex-1 (mTORC1) through phosphorylation of Raptor(Ser$^{792}$), (ii) de novo fatty acid synthesis through phosphorylation of acetyl-CoA carboxylase (ACC residue Ser$^{79}$), and (iii) the Raf-Mitogen Activated protein kinase (MAPK) pathway through Raf-Ser$^{621}$ phosphorylation, resulting in inhibition of anabolic events[19].

Canagliflozin is a sodium-glucose co-transporter-2 (SGLT2) inhibitor used to treat type 2 diabetes[20]. Inhibition of SGLT2 in the proximal nephron by canagliflozin blocks glucose reabsorption leading to increased glucose excretion[21]. Although canagliflozin improves glycemic control in diabetic patients, it does not appear to cause hypoglycemia in non-diabetics[22,23], where it has been shown to improve cardiovascular and renal outcomes[24]. The mechanisms mediating these protective effects of canagliflozin seem to be independent of its glucose lowering effects, are likely multifaceted and a subject of on-going debate. Our group was the first to illustrate that canagliflozin also suppresses mitochondrial complex-I and leads to activation of AMPK[25]. In subsequent studies, we showed that canagliflozin exerts anti-cancer activity in prostate and lung cancer cell lines[26], results subsequently verified by others[27].

In this study we, (i) examined canagliflozin's anti-tumor efficacy alone and in combination with radiotherapy in a variety of human cell and tumor models of PC and (ii) analyzed further the mechanism of action of this drug. We show that within its therapeutic window canagliflozin suppresses growth and enhances radiotherapy response in human androgen-sensitive and insensitive PC cells, and tumors, through suppression of the mTORC1-HIF-1$\alpha$ pathway. Canagliflozin modulates favorably multiple growth and survival pathways and mediates a marked reprogramming of PC transcriptional activity.

## Results and discussion

**Canagliflozin suppresses human PC proliferation, survival, and tumor growth, alone and in combination with radiotherapy.** PC develops as a heterogeneous multi-focal disease in the prostate and progresses in a similar fashion to accumulate mutations in a number of genes including androgen receptor (AR) and the tumor suppressors PTEN and Tp53[28]. Androgen/castrate-sensitive PC (CSPC) disease eventually evolves into castrate-resistant PC (CRPC)[29]. To enhance the applicability of our work, in this study we analyzed human castrate-sensitive (LNCap) and castrate-resistant PC (CRPC) cell lines that either express AR (22RV1) or not (PC3 and DU145). We began these studies with the evaluation of the anti-proliferative effects of canagliflozin and radiotherapy (Fig. 1a-c).

To ensure that our results reliably reflect canagliflozin's translational value for cancer therapy, we focused our in vitro work on concentrations of canagliflozin shown to be safely achievable in human serum[22,23]. Oral intake of canagliflozin (J&J Invokana) tablets of 100–300 mg (1.4–4.2 mg/kg/day in a 70 kg human) results in $C_{max}$ levels of 500–3500 ng/ml in patient plasma (1.1–7.9 $\mu$M)[30], while IV infusion of the drug is well tolerated up to concentrations of 38 $\mu$M[22]. For that, our in-vitro work focused on canagliflozin concentrations of 5–30 $\mu$M. Radiotherapy response was investigated in a variety of doses (2–8 Gy) but focused mostly on 2 Gy representing conventional or 4–6 Gy representing hypo- and ultra-hypo-fractionated radiotherapy used clinically in PC[31].

Figure 1a, b illustrates the response of PC cells to increasing doses of canagliflozin and radiation (RT). PC3 cells were most sensitive to canagliflozin compared to the other cell lines and LNCap and 22RV1 cells were more sensitive to radiotherapy, as described by others[32] (See Fig. 1a-b for IC$_{50}$ values and half inhibitory RT doses for each cell line). However, in combined treatments, canagliflozin significantly suppressed proliferation in irradiated cells, particularly at the lower radiotherapy doses used clinically (2–4 Gy, $p < 0.0001$) (Fig. 1c). To examine whether canagliflozin could help improve radiotherapy response in radioresistant PC cells, we analyzed DU145 cells developed by members of our group (DU145-RR), which were selected for survival after a long course of conventional-fractionation radiotherapy (2 Gy per day for 5 days a week for a total of 59 treatments)[33]. Canagliflozin provided improved growth suppression alone and in combination with radiotherapy (2 and 4 Gy) in DU145-RR compared to the parental controls (DU145) (Fig. 1c). Analysis of synergy demonstrated that canagliflozin and radiotherapy provide additive anti-proliferative activity in all PC cell lines (HSA scores:7.63–10.66)[34] (see Fig. s1a-e).

Clonogenic survival was also inhibited by canagliflozin in non-irradiated and irradiated cells (IC$_{50}$: 15.0, 16.62, 16.90 and

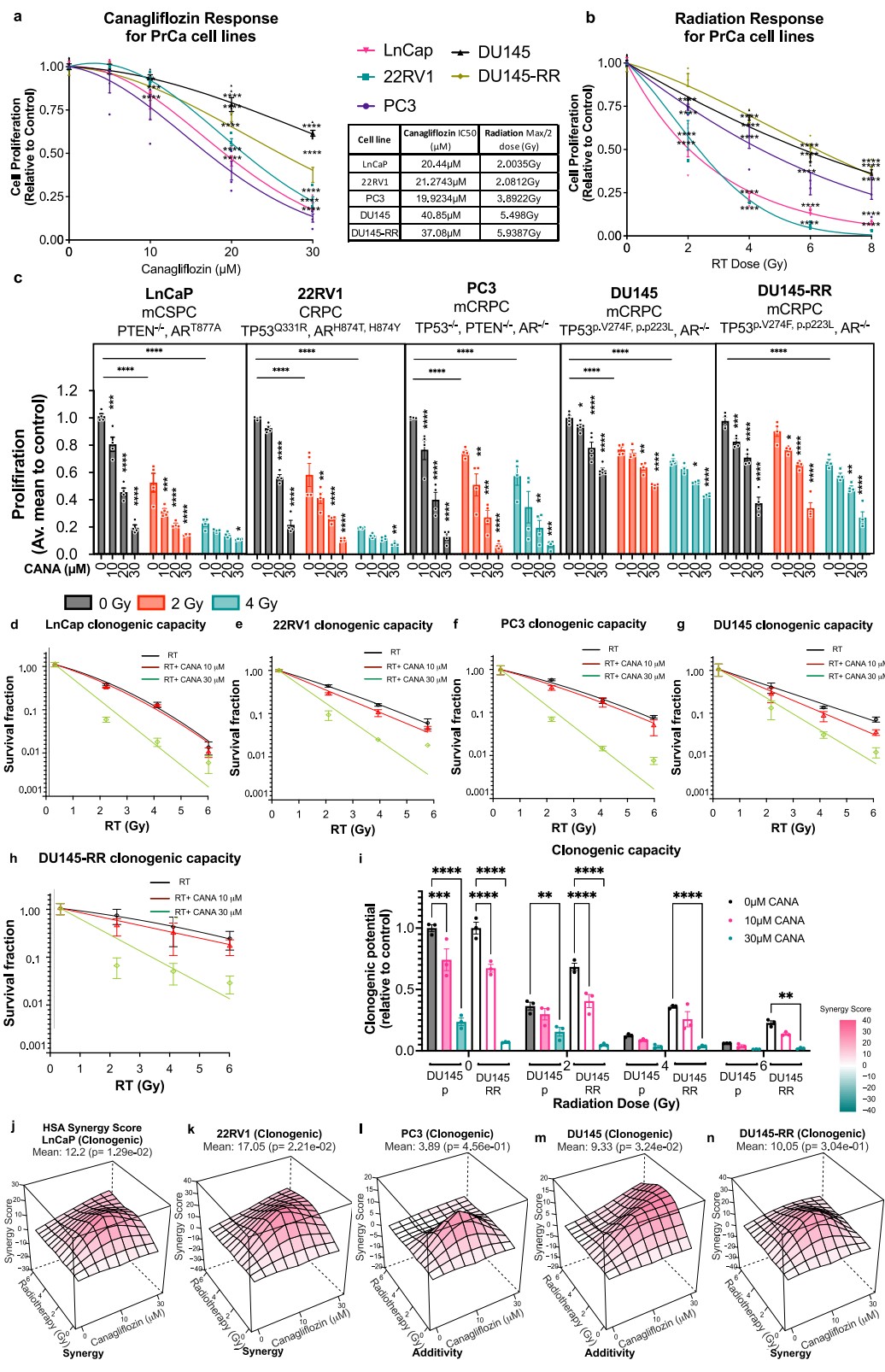

**Table (panel a):**

| Cell line | Canagliflozin IC50 (μM) | Radiation Max/2 dose (Gy) |
|---|---|---|
| LnCaP | 20.44μM | 2.0035Gy |
| 22RV1 | 21.2743μM | 2.0812Gy |
| PC3 | 19.9234μM | 3.8922Gy |
| DU145 | 40.85μM | 5.498Gy |
| DU145-RR | 37.08μM | 5.9387Gy |

20.81 μM, for PC3, 22RV1, LNCaP and DU145 cells, respectively, see Fig. s1f-g). However, irradiated 22RV1 and LNCaP cells (2–4 Gy) showed increased sensitivity to canagliflozin with reduced clonogenic IC$_{50}$ values (11.86 and 11.23 vs 14.9 and 15.2 μM, for 2 and 4 Gy, respectively, Fig. s1f-g). Fitting the data into the linear-quadratic model illustrated that at 30 μM canagliflozin provides sensitization of PC cells to radiotherapy

(Fig. 1d–h). Interestingly, untreated, and irradiated DU145-RR cells showed substantially improved response to the drug compared to parental cells (Fig. 1i). Analysis of synergy showed that canagliflozin and radiotherapy provide synergistic suppression of oncogenic potential in androgen-sensitive LNCaP and 22RV1 cells but have mostly additive efficacy in the remaining models (Fig. 1j–n).

**Fig. 1 Regulation of proliferation and clonogenic survival by Canagliflozin (CANA), radiation (RT) and combined treatment (CANA + RT). a** Effects of canagliflozin (CANA) or (**b**) radiotherapy (RT) on the cell proliferation of four prostate cancer (PC) cell lines with different mutation profiles (LnCaP, 22RV1, PC3, DU145) and one radio-resistant cell line (DU145-RR) treated with canagliflozin (CANA: 0-30 $\mu$M), or radiation (RT: 0-8 Gy). **c** Proliferation assays performed with PC cell lines treated with single agent or combined treatments, canagliflozin 0-30 $\mu$M and radiation (RT) (0-4 Gy), $n = 4$. **d**-**h** Clonogenic survival fraction data fitting into the linear-quadratic model was performed on all PC cell lines treated with canagliflozin (CANA) (0-30 $\mu$M) and radiation (RT) (0- Gy) as (**d**) LnCaP, (**e**) 22RV1, (**f**) PC3 (**g**), DU145, and (**h**) DU145-RR. **i** Clonogenic assay of DU145 compared to DU145-RR cells treated with canagliflozin (0–30 $\mu$M) and radiation (RT) (0–6 Gy), $n = 3$. **j**–**n** HSA synergy score was performed on all PC cell lines to determine synergism between radiation and canagliflozin based on clonogenic assay. The results were obtained by at least three separate experiments $n = 3$. The data show are mean ± SEM. Two-way analysis of variance (ANOVA) with the post hoc Tukey's multiple comparisons test was used to evaluate differences between the treatment and control groups. A two-way ANOVA was used to analyze comparison between parental DU145 and DU145-RR cell line, asterisks represent of p value, *$p < 0.05$, **$p < 0.01$, ***$p < 0.001$, and ****$p < 0.0001$. Micromolar ($\mu$M); Gray (Gy); Castration-sensitive prostate cancer (CSPC); Castration Resistant Prostate Cancer (CRPC); metastatic Castration Resistant Prostate Cancer (mCRPC); Radio-Resistant (RR); Androgen Receptor (AR).

Given the excellent bioavailability, pharmacokinetics, and safety profile of canagliflozin we subsequently conducted studies in mice. In these experiments, two different approaches were applied (Fig. 2a). PC3 cells were grafted in nude mice (lacking T-lymphocytes) and tumor growth kinetics were investigated, with the experimental endpoint reached for all animals when control animal tumors reached an average tumor size of 1200 mm$^3$. In contrast, 22RV1 cells were injected into NRG mice (lacking T- and B-lymphocytes), and that experiment was designed with individual animal endpoints set when the tumor reached 2200–2500 mm$^3$. Addition of canagliflozin to chow diet (45–70 mg/kg/day, see Methods) cause increased food consumption in NRG mice grafted with 22RV1 tumors but did not affect food intake in nude mice grafted with PC3 tumors (Fig. 2b) or the animal whole body weight in either model (Fig. s2a). As expected, given the effects of the drug to induce diuresis, canagliflozin treatment increased water consumption in both models (Fig. 2c).

In these experiments canagliflozin was compared to and added to radiotherapy (5 Gy), a treatment we established earlier to mediate 50% inhibition of tumor growth, a desirable effect in drug combination experiments. Canagliflozin suppressed growth kinetics of non-irradiated PC3 tumors (45% inhibition at endpoint), an effect similar to that mediated by radiotherapy. The addition of the drug in animals with irradiated tumors further amplified the effects of radiotherapy, seen in both in vivo (Fig. 2d) and ex vivo volume and weight measurements (Fig. s2b, c). Canagliflozin was equally effective in 22RV1 tumors (Figs. 2e, f, s2d, e). Tumors in 22RV1 xenograft drug-treated animals reached endpoint in 30 days, compared to 18 days in the control group (12-day delay) (Fig. 2e, f). Inhibition of tumor growth by radiotherapy paralleled that of canagliflozin (endpoint at ~30 days). However, combined treatment dramatically slowed tumor growth further (endpoint at ~50 days). Kaplan–Meier survival curves illustrate the impact canagliflozin could provide on survival alone ($p = 1 \times 10^{-5}$) and in combination with radiotherapy ($p = 2 \times 10^{-6}$) (Fig. 2f).

To understand the mechanism of tumor suppression, we examined the tumors for markers of DNA replication, apoptosis, and necrosis. Histone-H3 phosphorylation (P-H3(Ser$^{10}$)) immunohistochemistry (IHC) reliably identifies cells undergoing DNA replication and mitosis in PC3 and 22RV1 tumors. Canagliflozin suppressed P-H3(Ser$^{10}$) in both non-irradiated and irradiated PC3 and 22RV1 tumors, while combined treatment caused a small additional suppression in both types of xenografts (Fig. 2g, h). Tumor necrosis analysis performed digitally (Fig. s2f) demonstrated increased necrosis with both canagliflozin and radiotherapy treatments but no additive effects of combined treatment (Fig. s2g). In contrast, canagliflozin and radiotherapy enhanced the levels of the apoptosis marker phosphorylated cleaved-caspase-3

P-CC3(Asp$^{175}$) and combined treatment induced further increase in 22RV1 tumors (Fig. 2i–j).

The observations made to this point provide strong evidence that canagliflozin can indeed suppress proliferation, oncogenic potential, and tumor progression of both androgen-sensitive and -insensitive PC models alone and in combination with radiotherapy. In vivo these effects were mediated with oral delivery of clinical grade canagliflozin (Invokana tablets in diet). Studies with xenograft models of hepatocellular or thyroid cancer detected anti-tumor activity with oral canagliflozin delivery at 50–300 mg/kg/day[35,36]. In a genetically diverse mouse cohort, Miller et al. [37]. observed improved metabolic parameters and longevity with long-term canagliflozin treatment in diet at 30 mg/kg/day, resulting in 0.36–0.86 mg/ml or 0.8–1.9 $\mu$M in plasma. Based on that, we doubled the drug concentration in diet (calculated to achieve approximately 60 mg/kg/day). Diet consumption assessments indicated that our animal cohorts received orally an estimated 40–70 mg/kg/day (see Methods), suggesting probable achievement of a drug bioavailability that would approach that detected in patients receiving canagliflozin orally (Invokana tablets 100–300 mg, generating to 1.1–7.9 $\mu$M in plasma)[30]. This treatment induced metabolic stress signals (P-ACC(Ser$^{79}$)), inhibited DNA replication (P-H3(Ser$^{10}$)) and induced apoptosis (increased P-CC3(Asp$^{175}$)) markers, and significantly ($p < 0.0001$) suppressed tumor growth in PC tumor tissue, with no evidence of physical stress or weight loss in animals. (Fig. 2).

**Canagliflozin mediates significant reprogramming of PC transcriptional activity.** To better understand canagliflozin's anti-tumor mechanism, we performed RNAseq analysis on PC3 cells treated with 10 $\mu$M canagliflozin alone, a concentration within its therapeutic window[22]. PC3 cells were selected for these studies, a model of metastatic CRPC that lacks AR expression. Canagliflozin treatment for 24 h led to statistically significant downregulation of 929 and upregulation of 1324 genes in PC3 cells (FDRq<0.05) (Fig. 3a). Figure 3b illustrates the dichotomy of transcriptional regulation by canagliflozin, highlighted by the most significantly downregulated (TRIM28 and TXNIP) and upregulated (TFPI and DUSP5) genes (FDRq = $2.4 \times 10^{-21}$, $1.5 \times 10^{-17}$, $1.1 \times 10^{-34}$, $2.3 \times 10^{-27}$, respectively). TRIM28 is an E3-ubiquitin ligase effector substrate that enhances androgen receptor signaling[38], whereas TXNIP is a thioredoxin-binding protein that regulates cellular redox signaling and ER-stress[39]. On the other hand, DUSP5 is a nuclear bispecific MAPK phosphatase that suppresses nuclear MAPK signaling[40]. Tissue factor pathway inhibitor (TFPI), a gene which was upregulated and encodes a negative regulator of hemostasis, was the most significantly upregulated gene. Its potential role in canagliflozin's action is unclear and should be investigated in future studies.

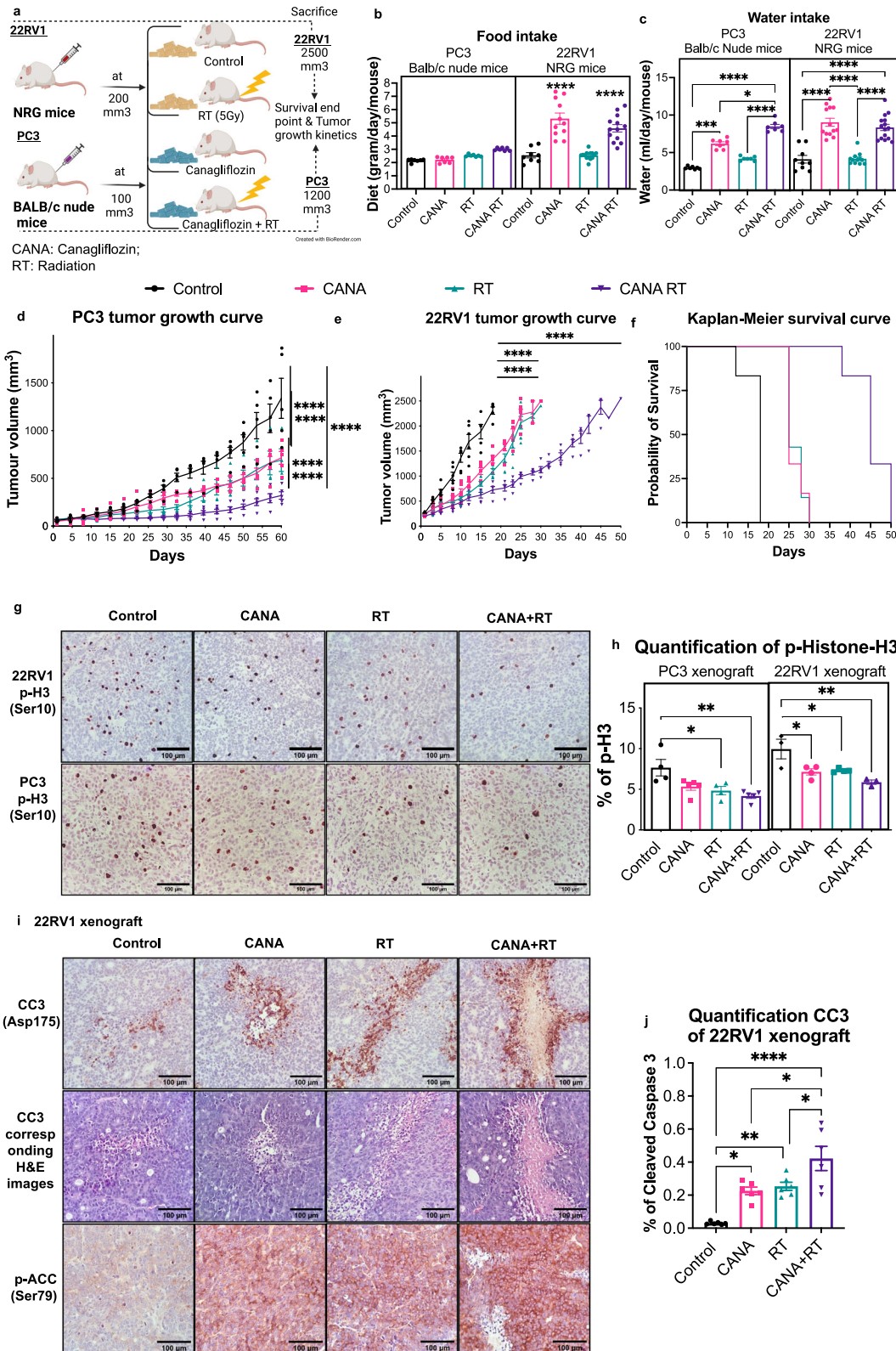

Gene ontology (GO) analysis performed with the Gene Set Enrichment Analysis (GSEA) showed that 151 Gene Ontology Biological Processes (GOBP) were significantly upregulated and 55 downregulated by canagliflozin. Examples of biological processes regulated significantly by canagliflozin alone include: i) downregulation of gene sets related to the Wnt pathway, Aurora and Polo kinases, angiogenesis, cell cycle regulation, growth factor (e.g., EGF, IGF-I, Insulin), Ras family small GTP-binding protein signaling pathway, involving MAPK pathways, E2F, and others; ii) upregulation of eukaryotic translation, oxidative phosphorylation (OxPhos), TCA cycle, ATP synthesis, mitochondrial function, ribosome biogenesis and immune response-related genes (Fig. 3c, see supplementary data file 1.xlsx for complete list).

**Fig. 2 Efficacy of canagliflozin (CANA) in combination with radiotherapy (RT) in vivo. a** Schematic of NRG and BALB/C nude mice xenograft experimental design. **b** Food intake (gram/mouse/day) and (**c**) water intake (mL/day/mouse). **d** Tumor growth kinetics in PC3 xenograft BALB/C nude mice: four groups (n = 5 per group), two-way ANOVA, asterisks represent of p value, *p < 0.05, **p < 0.01, ***p < 0.001, and ****p < 0.0001. **e–f** Tumor growth kinetics and Kaplan–Meier survival curves in NRG mice 22RV1 xenografts: four groups (n = 6 per group), two-way ANOVA with Tukey's multiple comparison for (**e**) and log-rank (Mantel–Cox) tests for (**f**), ****P < 0.0001 for CANA, RT, and CANA + RT compared to control. Log-rank test for trend was found for the CANA + RT group to control, ****P < 0.0001. **g** Representative images of immunohistochemistry (IHC) analysis for phosphorylated-histone3 (P-H3(Ser[10])) antibody of tumors from PC3 grafted in nude mice and 22RV1 grafted in NRG mice and (**h**) its IHC quantification for p-H3(Ser[10]). **i** Representative images of IHC examination of H&E staining, phosphorylated-ACC(Ser[79]), and phosphorylated cleaved-caspase3 (P-CC3(Asp[175]) antibody of tumors from 22RV1 NRG mice. **j** IHC quantification for P-CC3(Asp[175]). Ordinary one-way ANOVA with the post hoc Bonferroni's multiple comparisons test and repeated measures were used to evaluate differences between groups, asterisks represent of p value, *p < 0.05, **p < 0.01, ***p < 0.001, and ****p < 0.0001. Data shown as Mean ± SEM.

RNAseq analysis revealed that canagliflozin at clinical achievable dose (10 μM), demonstrated dual effects on gene expression in PC3 cells. It mediated feedback upregulation of genes related to autophagy, organelle acidification V-ATPase (ATP6V-0E1, −1D, −1E1, −1G1, −1H), and genes encoding subunits of mitochondria respiratory chain enzymes [complex-I: (ND1, NDUFA1-FV2), complex-II (SDHB-D) and, complex-III (COX1-8A, UQCRB-RQ)] (Fig. 3d). In contrast, canagliflozin downregulated genes involved in the MAPK-H3 and mTORC1-p70S6k/4EBP1-related genes, including MAPK-Kinase-Kinase-11 (MAP3K11; an activator of B-Raf, Jun-N Terminal-kinase (JNK), Erk and p38 MAPK), MAPKs p38-alpha (MAPK14), and MAPK-Activated-Protein-kinase-3 (MAPKAPK3: a target of Erk1/2), Akt1, Akt2, ribosomal p90-S6-kinase (RPS6KA1) and PI3k regulatory (PIK3R1, PIK3R3) and catalytic (PI3KC3) subunits (Fig. 3e). Canagliflozin's effects extended to regulating the expression of early signaling mediators and effectors of tyrosine kinase receptor pathways upstream of MAPKs and Akt, such as adaptor molecules IRS, SHC3 and GRB2, as well as small GTP-binding proteins involved in signal transduction (RAC2) and transcription factors such as EGR2, REST, ELK1, and E2F (including E2F2, E2F3, and E2F4).

Interestingly, we found that canagliflozin downregulated transcription of genes that enhance HIF-1α degradation such as Protein-Kinase C-alpha (PRKCA)[41]. This was associated with downregulation of genes that are directly regulated by HIF-1α, such as VEGF and TGFB3[12,42] (Fig. 3f). Moreover, the drug increased expression of ELOB, ELOC, RBX1, and VBP1 which participate in the formation of the VHL-box E3-ubiquitin ligase that targets HIF-1α for degradation[43].

Furthermore, canagliflozin mediated substantial modulation of key genes involved in DNA replication and cell cycle progression. Gene Ontology DEG sets of cell cycle phase transition (GO:0044770) and cell cycle progression (GO:0051726) were suppressed by canagliflozin. These included genes involved in G1-S transition (CDC25A, MCMC2, CDK2 and CDC20), G2-M checkpoint (GTSE1, CCNF and ATF5), M-phase and overall cell cycle and DNA replication regulators (PLK1, E2F transcription factors, CDCA8, POLD1, TIMELESS), and sensors of cellular stress that actively regulate cell cycle (MAPK14 and GADD45B) (Fig. 3g).

These results illustrated the substantial complexity and involvement of multiple biological processes in canagliflozin's mechanism of action. They demonstrate that within clinically achievable doses canagliflozin alone is able to suppress key molecular pathways supporting survival and tumor progression.

**Canagliflozin inhibits mitochondrial OxPhos and induces a metabolic stress response.** Studies suggested that canagliflozin's anti-tumor activity relies on glucose-uptake through SGLT2 in liver cancer[27]. However, other studies found that the anti-proliferative effects of canagliflozin are not influenced by glucose-update or the level of SGLT2 expression[44]. Papadopoli et al.[44],

found that Canagliflozin is a potent inhibitor of mitochondrial respiration and total ATP production in human breast cancer cells. In their study, Papadopoli et al. [44], did not detect complex-I inhibition in isolated mitochondria derived from murine tissue, but rather complex-II, indicating a potential differential sensitivity of complex I to canagliflozin between mammalian species. Further, they detected inhibition of glutamine metabolism in breast cancer in response to the drug[44]. Nevertheless, we have shown that extracellular levels of glucose or pyruvate supplementation do not reverse canagliflozin's anti-tumor activity in PC, while drug action is inhibited by the expression of biguanide insensitive *Saccharomyces cerevisiae* complex-I core subunit NADH dehydrogenase (ND1)[26], demonstrating the dependence of canagliflozin action on complex-I activity. Further, the work of our group and others showed that canagliflozin inhibits specifically complex-I but not complex II-IV[45].

Based on those earlier observations, we examined next the effect of the drug on respiration of untreated and irradiated PC cells. Canagliflozin suppressed significantly (p < 0.0001) the basal and maximal oxygen consumption rate (OCR) as well as the estimated mitochondria-linked ATP production, in non-irradiated and irradiated cells. Figure 4a–c illustrates data normalized for cellular content (see Fig. s3a-b for raw OCR curves). Radiotherapy significantly (p < 0.0001) enhanced basal and maximal OCR and estimated mitochondria-linked ATP production (at 8 Gy), but canagliflozin was able to effectively abrogate these events (Fig. 4a–c). Consistent with increased OCR, radiotherapy appeared to suppress basal extracellular acidification (ECAR) rates. However, despite blockade of mitochondrial respiration, canagliflozin did not alter ECAR in either control or irradiated cells (Fig. 4d). The OCR/ECAR ratio, a marker of metabolic phenotype, was enhanced by radiotherapy but this was blocked by canagliflozin (Fig. 4e).

The inhibition of OCR observed in response to canagliflozin treatment is likely responsible for the induction of genes encoding subunits of mitochondria respiratory chain enzymes and part of a feedback response (Fig. 3d). On the other hand, the lack of significant change in ECAR rate is consistent with the lack of upregulation of glycolysis related genes (Fig. s3c). Interestingly, canagliflozin suppressed the expression of a number of transporter genes including SLC16A14, a gene believed to facilitate transport of H$^+$ and monocarboxylates, like lactate, across the plasma membrane[46] (Fig. s3c).

To determine whether canagliflozin mediates its action solely through OxPhos inhibition, we compared its anti-proliferative activity to that of two specific OxPhos complex-I inhibitors, BAY-87-2243[47] and IACS-010759[48]. At widely used doses, these agents significantly suppressed OCR in glycolytic rate bio-analyzer assays (Fig. s3e). However, unlike canagliflozin, BAY-87-2243 and IACS-010759 significantly increased the proton efflux rate (PER) (Fig. 4f). Despite more effective suppression of OCR, we observed a limited 30-40% inhibition of PC3 cell proliferation by these agents, compared to 45% and 80% with 10 or 30 μM

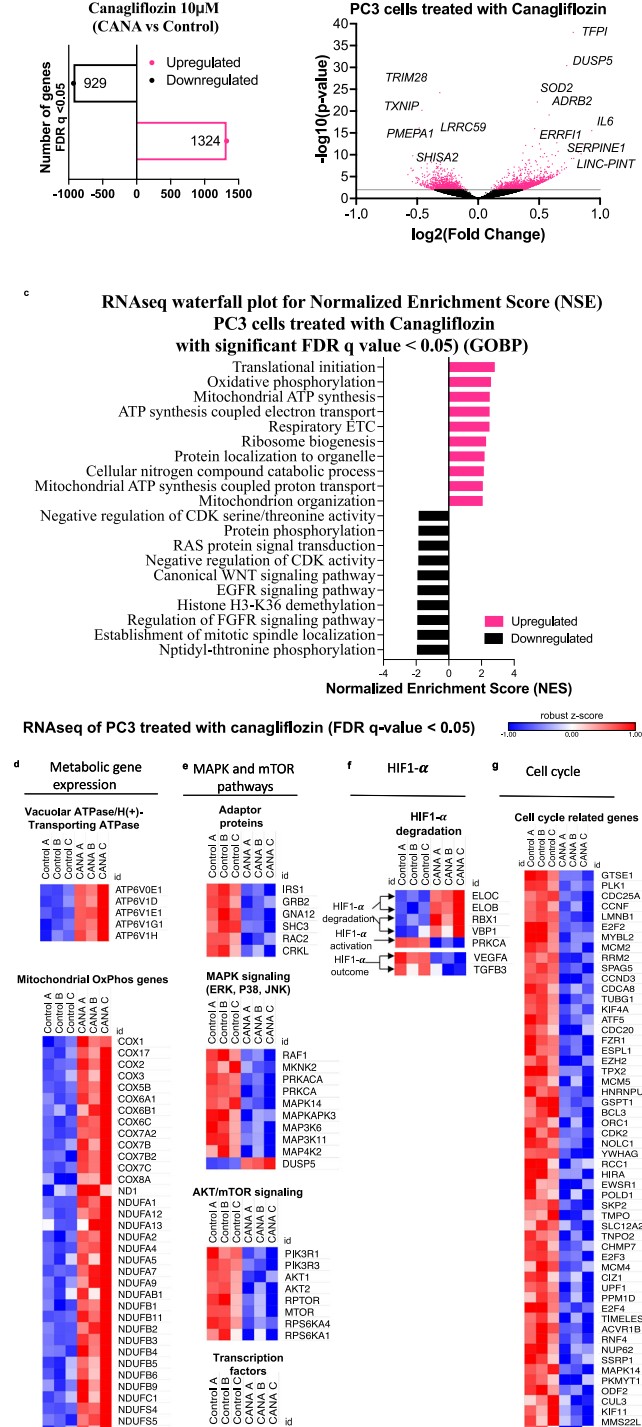

**Fig. 3 RNAseq analysis in PC3 cells treated with canagliflozin. a** Number of significant upregulated and downregulated genes by 10 $\mu$M canagliflozin with False Discovery Rate FDR q-value < 0.05. **b** A volcano plot illustrates the total number of genes that are up- and down-regulated by canagliflozin, as well as the top regulated genes identified from PC3 RNAseq datasets. All significant genes are represented by red dots, whereas genes with p-values larger than 2 ($-\log10(p\text{-}value)$) are separated by a gray line and have an FDR q-value < 0.05. Positive log2 fold change indicating upregulated genes and negative indicating downregulated genes. **c** Waterfall plot from RNAseq analysis, depicts the most important biological processes controlled by canagliflozin, including the top 10 downregulated and upregulated Gene Ontology Biological Process (GOBP) pathways. The differentially expressed genes (DEGs) are with FDR q-value < 0.05. **d–g** Heatmap of RNAseq data (analyzed in triplicates A, B, C) illustrating differentially gene expression in PC3 cells after 24 h of treatment with 10 $\mu$M canagliflozin with significant False Discovery Rate (FDR) q-value of > 0.05. Expression of genes related to (**d**) mitochondrial OxPhos, and organelle acidification V-ATPase, (**e**) MAPK, AKT/mTOR pathway (including the adaptor upstream genes and downstream transcription factors, (**f**) HIF1-$\alpha$ degradation-related genes, and (**g**) cell cycle phase transition and cell cycle progression. Heatmap gene expression is represented by normalized log2 feature-counts (with robust z-score). The blue to red scale represents downregulated to upregulated from −1 to 1.

target of AMPK inhibiting ACC (Figs. 4h and s3d). Total ACC and P-ACC(Ser[79]) levels diminished in cultured irradiated cells over 24 h. However, in xenografts a single fraction of radiotherapy and daily animal treatment with canagliflozin caused chronic induction of P-ACC(Ser[79]) and combined treatment increased this more consistently (Fig. 2i). These findings demonstrate canagliflozin's ability to suppress de novo lipogenesis in untreated and irradiated PC cells and tumors, a pathway vital to cancer growth and survival.

Taken together, this work shows that radiotherapy enhances OCR and reliance on oxidative metabolism (OCR/ECAR ratio) in PC cells but canagliflozin blocks OCR mediating a mild metabolic stress state. This is associated with the activation of metabolic stress signaling (AMPK) and feedback induction of OxPhos genes. Unlike, the investigational specific OxPhos inhibitors (BAY-87-2243 and IACS-010759) canagliflozin inhibits OxPhos without increasing extracellular acidification/proton efflux rates or glycolytic gene expression. This finding is indeed intriguing given the established role of glycolysis in cancer cell survival. Canagliflozin has additive anti-proliferative efficacy over specific OxPhos inhibitors indicating that this drug engages additional molecular pathways in its anti-proliferative action beyond metabolic stress signals.

The enhancement of PC cell mitochondrial respiration by radiotherapy we observed here is consistent with our previous reports[50] and work of other groups indicating a potential dependence of irradiated cells on mitochondrial respiration[51]. This could result in a specific sensitivity of irradiated cells to mitochondrial inhibitors and the generation of a niche for synthetic lethality when OxPhos inhibitors are combined with radiotherapy.

Activation of AMPK and inhibition of ACC could contribute to canagliflozin's tumor suppressor activity. However, earlier we found that canagliflozin's anti-proliferative activity remained intact in cells expressing dual knock-in of ACC lacking the AMPK regulatory phosphorylation sites Ser[79/212]/Ala, suggesting suppression of cell growth by canagliflozin is not dependent on inhibitory regulation of ACC by AMPK[26]. Further, expression of dominant-negative AMPK-$\alpha$1-subunit, or $\beta$1-subunit knockout, mediated minimal reversal of canagliflozin's anti-proliferative action[26]. Nevertheless, given that PC3 cells also express AMPK-$\alpha$2- and $\beta$2 subunits[49] that can support assembly of functional

canagliflozin, respectively. Further, canagliflozin induced additional anti-proliferative efficacy when combined with BAY-87-2243 and IACS-010759 (Fig. 4g).

We hypothesized that metabolic stress induced by canagliflozin mediates activation of AMPK in both untreated and irradiated cells. As we observed earlier[49], radiotherapy increased the total AMPK-$\alpha$ subunit levels over the first 24-30 h post-treatment. Canagliflozin increased and activated P-AMPK-$\alpha$(Thr[172]). This activation was associated with increased P-ACC(Ser[79]), a specific

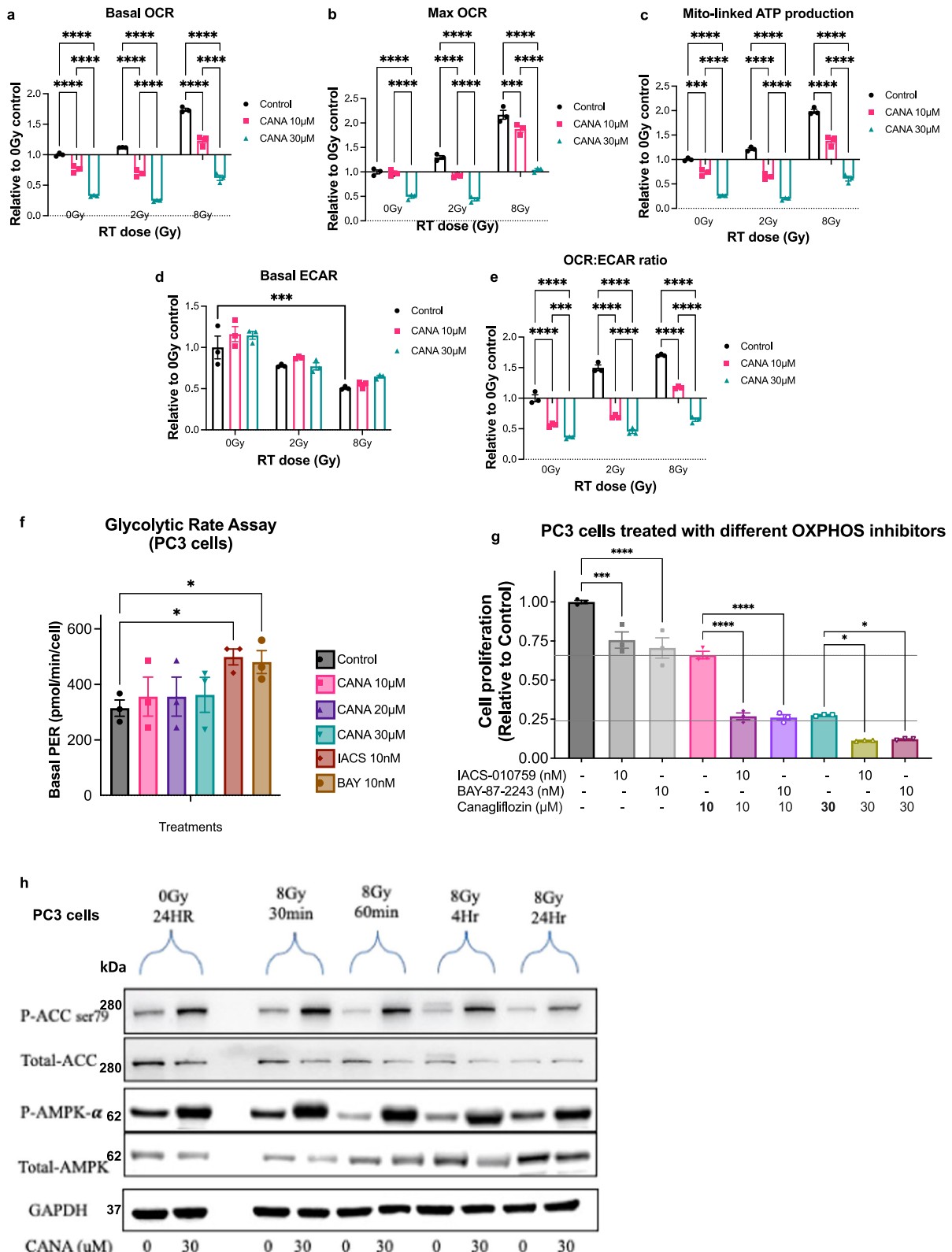

AMPK-α2/β2/γ1−2 complexes, AMPK heterotrimers may still be involved in canagliflozin's anti-tumor activity.

**Canagliflozin blocks growth factor signal transduction**. In PC cells, canagliflozin induced sustained inhibitory P-Raptor(Ser[792]), a specific AMPK target that inhibits mTORC1[52]. Interestingly,

P-Akt(Ser[473]), a target of the mTORC2 complex, involved in the activation of this enzyme, was inhibited by canagliflozin, without a consistent effect on the P-Akt(Thr[308]). Further, the drug blocked P-mTOR(Ser[2448]), a target of Akt[53] and reduced or abolished activating phosphorylation events of mTORC1 effectors including p70[S6k], S6 and 4EBP1 in both non-irradiated (control) and irradiated cells (2 and 8 Gy), (Fig. 5a, b). These results

**Fig. 4 Regulation of mitochondrial respiration by canagliflozin (CANA) in PC3 cells.** Effects of canagliflozin and radiation (RT) treatments on (**a**) basal OCR, (**b**) maximum OCR, (**c**) mitochondrial linked-ATP production, (**d**) basal ECAR, and (**e**) OCR:ECAR ratio. The data were normalized to cell content, and then expressed as a mean average to 0 Gy control. A two-way ANOVA with the post hoc Tukey's multiple comparisons test was used to determine the significant difference between groups, asterisks represent of p value, *$p < 0.05$, **$p < 0.01$, ***$p < 0.001$, and ****$p < 0.0001$. **f** Glycolytic challenge basal proton efflux rate (PER) and (**g**) cell proliferation assay for canagliflozin compared to other two OxPhos complex-I inhibitors, and IACS-010759. Ordinary one-way ANOVA with the post hoc Bonferroni's multiple comparisons test was performed, to determine if there were significant changes between the treatment and control groups. Activation of metabolic stress pathways. **h** Immunoblotting assay for ACC (total), phosphorylated-ACC(Ser[79]), AMPK-α(total), and phosphorylated AMPK-α(Thr[172]). The vertically stacked strips of bands presented in the figure panels are not all derived from a single membrane. The significance of the time course experiment at each treatment dosage was determined using a two-way analysis of variance (ANOVA), asterisks represent of p value, *$p < 0.05$, and **$p < 0.01$. The data show mean ± SEM; $n = 3$.

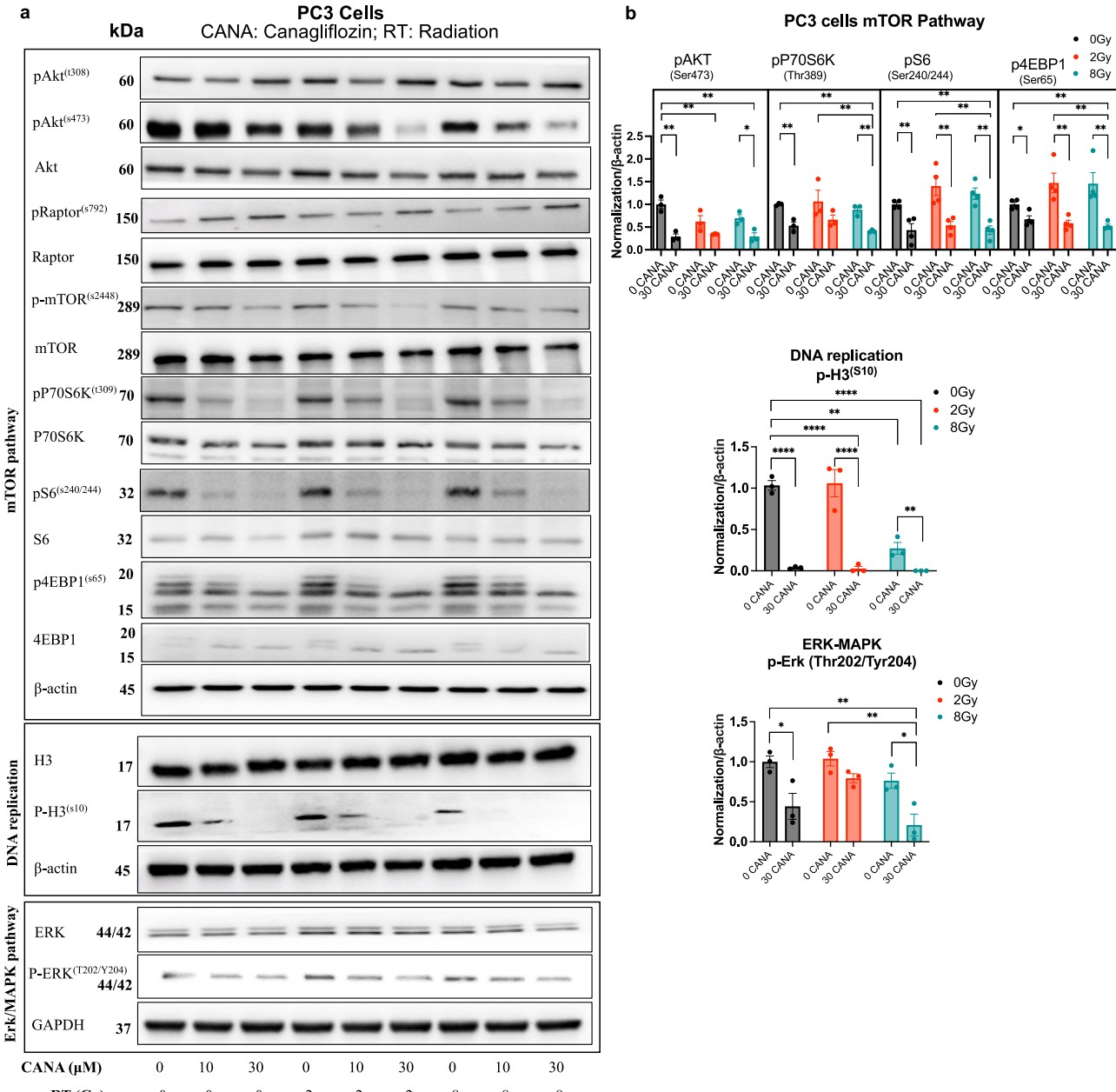

**Fig. 5 Modulation of the Akt/mTOR-p70$^{S6k}$/4EBP1, DNA replication and MAPK pathways. a** Several markers of the mTOR, DNA replication, and MAPK pathways were immunoblotted in PC3 cells treated with canagliflozin (CANA) (10–30 μM), radiation (RT) (0–8 Gy), or the combination (CANA + RT). **b** Immunoblotting quantification of phosphorylated protein levels and normalized to β-actin. Data are shown as Mean ± SEM; $n = 3$. The vertically stacked strips of bands presented in the figure panels are not all derived from a single membrane. A two-way ANOVA with post hoc Tukey's multiple comparisons test was used for comparisons, asterisks represent of p value, *$p < 0.05$, **$p < 0.01$, ***$p < 0.001$, and ****$p < 0.0001$.

highlight canagliflozin's potential to suppress the vital mTOR-protein synthesis axis in PC cells.

P-H3(Ser[10]) is a well-described target of extracellular signal-regulated kinase (Erk)-MAPK[54]. Canagliflozin suppressed P-H3(Ser[10]) in PC3 cells without affecting total H3 levels (Fig. 5a, b, $p < 0.0001$). Radiation reduced P-H3(Ser[10]) levels but canagliflozin suppressed P-H3(Ser[10]) levels even further in irradiated cells. This is consistent with the results of (Fig. 2g, h) illustrating that in PC3 and 22RV1 tumors, canagliflozin in combination with radiotherapy significantly decreased P-H3(Ser[10]). The drug showed similar efficacy in suppressing P-Erk1/2(Thr[202]/Tyr[204]) in control and irradiated cells, which are recognized activating phosphorylation sites on the two kinases by upstream effectors[55], (Fig. 5a, b). These results are congruent with the upregulation of *DUSP5* by canagliflozin, a nuclear MAPK phosphatase and the observed downregulation of MAPK-, mTORC1- related genes and upstream adaptor molecules (Fig. 3e).

The observed findings suggest that canagliflozin exerts wide inhibitory control on both transcriptional and post-translational events in growth factor signal transduction pathways.

**Canagliflozin regulates PC growth through HIF-1α suppression.** HIF-1α is a key regulator of glycolysis, survival, and radio-resistance pathways in cancer cells[14,16]. Cellular HIF-1α levels normally remain low during normoxia through hydroxylation, which flags HIF-1α for proteasome degradation initiated by the E3-ubiquitin ligase complex[43]. Since, the Akt-mTORC1 pathway regulates HIF-1α translation[12], and our RNAseq interestingly determined that canagliflozin upregulated transcription of genes that enhance HIF-1α degradation via induction of E3-ubiquitin ligase complex related genes, we analyzed HIF-1α protein levels after drug and radiotherapy treatment. Canagliflozin suppressed HIF-1α levels in control PC cells ($p < 0.0001$) and helped reduce its levels further in irradiated cells ($p < 0.0001$) (Fig. 6a, b). Overall, our results suggest a dual mechanism by canagliflozin to eliminate HIF-1α levels through, i) suppression of synthesis by inhibition of mTOR-protein synthesis axis and ii) enhancement of degradation via ubiquitination through the E3-ubiquitin ligase complex, which is not directly regulated by AMPK and mTORC1. This dual mechanism can have far reaching effects on tumor progression and survival.

To address the significance of HIF-1α regulation by canagliflozin in the mechanism of drug action, we examined canagliflozin in combination with the clinically used prolyl-hydroxylase (PDH) inhibitor Roxadustat[56], which blocks HIF-1α degradation. Canagliflozin reduced HIF-1α levels but Roxadustat (5–30 µM) enhanced baseline levels of HIF-1α in PC3 cells and prevented HIF-1α downregulation in canagliflozin-treated cells (Fig. 6c). Roxadustat mediated partial but significant reversal (25%, $p < 0.0001$) of canagliflozin-mediated inhibition of proliferation in cells treated with combined therapy (Canagliflozin+Roxadustat) (Fig. 6d).

These results show that HIF-1α suppression is indeed involved in the mechanism of canagliflozin's anti-tumor activity. These findings agree with observations in hepatocellular carcinoma models[36].

**Regulation of cell cycle by canagliflozin.** Based on our RNAseq results, we observed a significant downregulation of genes involved in cell cycle progression. Considering these findings, we examined canagliflozin's effects on cell cycle and cycle checkpoint regulators. Immunoblotting of PC3 lysates showed that canagliflozin induces the expression of cyclin-dependent kinase inhibitors p27[kip1] and p21[cip1], mediating G1/S and G2/M checkpoints (Fig. 7a). For that, we analyzed the cell cycle with flow cytometry. Canagliflozin did not significantly change the cell cycle distribution of non-irradiated PC3 cells but led to G1 phase arrest and

reduction in cells found at the G2/M checkpoint (Fig. 7b). Interestingly, canagliflozin triggered G1 arrest ($p = 0.0423$) and decreased S phase cells in non-irradiated 22RV1 cells but also caused significant redistribution of cells in G2/M ($p = 0.0037$) and reduced S1 phase cells in irradiated 22RV1 (Fig. 7c). In agreement with these findings, canagliflozin increased p21[cip1] levels in 22RV1 cells, which was potentiated in irradiated 22RV1 cells (Fig. 7d) compared to PC3 (Fig. 7a). We examined whether the canagliflozin-mediated regulation of p21[cip1] was associated with events leading to p53 stabilization and activity, such as p53 phosphorylation on P-p53(Ser[15]), induced by ATM in response to DNA damage. Radiotherapy induced rapid P-p53(Ser[15]) and maintained this effect long-term, but canagliflozin did not alter the levels of Ser[15] phosphorylation on p53 over this time course (Fig. 7d).

The differential regulation of the cell cycle by canagliflozin in PC3 vs 22RV1 cells may be related to a lack of TP53 activity in the former line, which mediates cell cycle checkpoints in response to Radiotherapy. The G2/M checkpoint is a well described radio-sensitive phase of the cell cycle[57]. The observed increased redistribution of 22RV1 cells into the G2/M phase correlates well with detection of improved synergy between canagliflozin and radiotherapy in suppressing clonogenic survival in 22RV1 cells compared to observed additivity in PC3 cells (HAS scores 17.05 vs 3.89, respectively, Fig. 1k–l). This indicates a potential for canagliflozin to provide radio-sensitizing activity in cells with intact p53 activity. Nevertheless, differential regulation of the cell cycle by canagliflozin did not translate to significantly different anti-proliferative and tumor suppressive efficacy in irradiated tissue cultures and xenografts of the two models.

Based on these findings we conclude that cell cycle regulation is unlikely to be a key pathway in mediating drug action when canagliflozin is combined with radiotherapy.

**Transcriptional reprogramming in irradiated cells: Canagliflozin induces mitochondria metabolism and p53 pathway and suppresses the cell cycle progression and DNA replication genes.** The above observations indicated the value of understanding canagliflozin's efficacy in regulating gene expression in irradiated cells. We performed another RNAseq analysis with 22RV1 cells, a radio-sensitive PC line (see Fig. 1), 24 h after canagliflozin (10 µM), radiotherapy (5 Gy) or combined treatment. The number of significantly upregulated vs downregulated genes by canagliflozin, radiotherapy and combined treatment, were 135 vs 92, 4159 vs 4348 and 4215 vs 4360 compared to control, respectively. A total of 982 genes were differentially downregulated and 939 genes were upregulated in irradiated cells with the addition of canagliflozin (Fig. 8a, see Fig. s4a: for a complete description of data including results on transcript levels in untreated and canagliflozin alone treated cells). The volcano plot (Fig. 8b) illustrates DEGs in response to canagliflozin treatment in irradiated 22RV1 cells (FDRq<0.05). The cytokine growth and differentiation factor-15 (*GDF15*), the ribosomal protein S17 (*RPS17*) and the p21[cip1] gene (*CDKN1A*) were the most significantly upregulated genes by canagliflozin in irradiated cells (FDRq=$5.5 \times 10^{-17}$, $2.7 \times 10^{-14}$, $2.3 \times 10^{-11}$, respectively). On the other hand, *RRM2*, a Ribonucleotide Reductase Regulatory Member M2 gene that supports DNA synthesis, and *MCM4*, a subunit of the crucial replicative helicase MCM complex that unwinds DNA forks during DNA replication, were the most significantly downregulated genes by canagliflozin in irradiated cells (FDRq=$8.3 \times 10^{-15}$, $8.9 \times 10^{-12}$, respectively). GO analysis (using GSEA) showed that 155 GO biological pathways were upregulated and 132 were downregulated significantly (FDRq < 0.05) by canagliflozin in irradiated cells. Significant

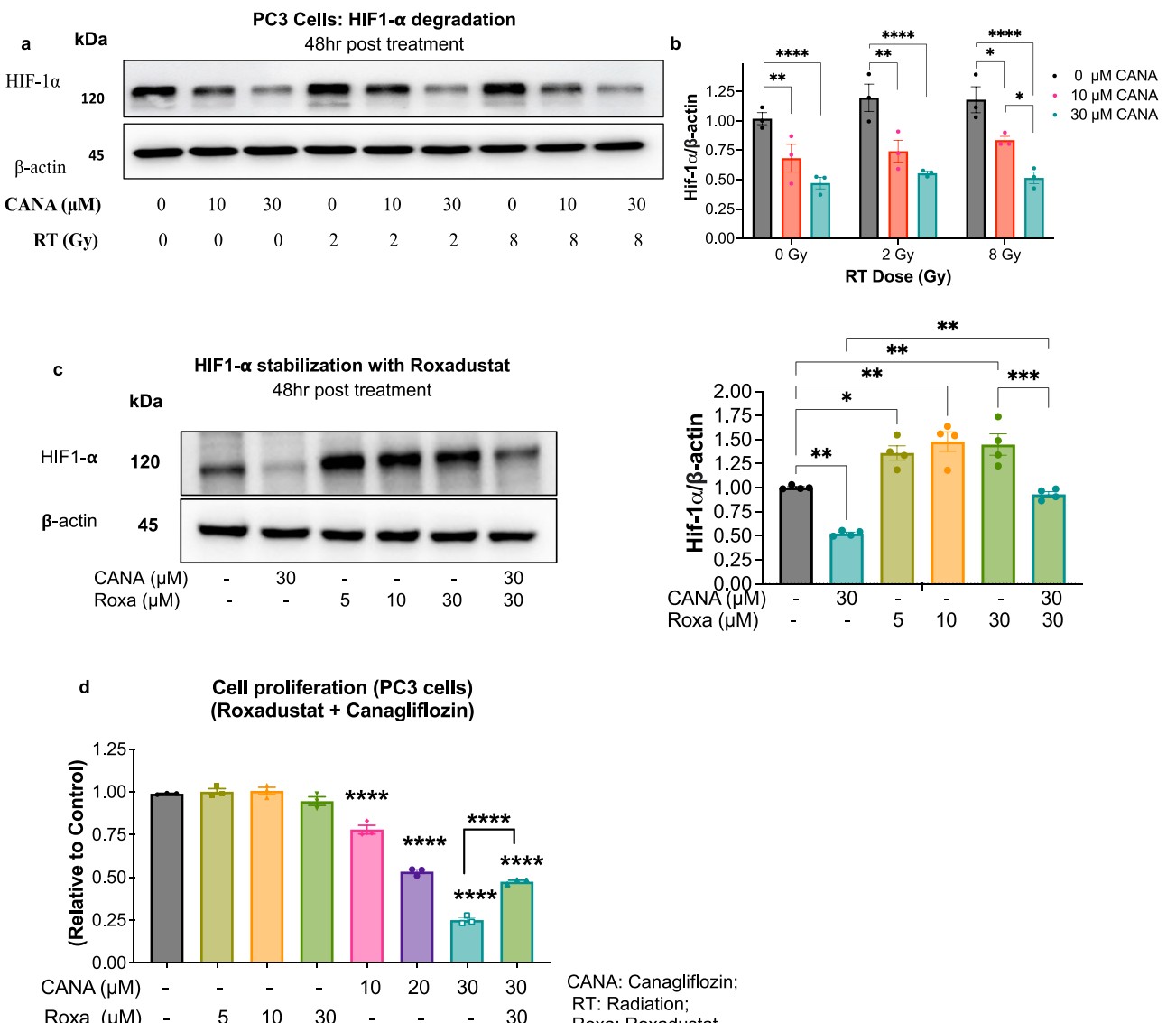

**Fig. 6 Regulation of HIF1 pathway in prostate cancer (PC). a** Immunoblotting assay for HIF-1α in PC3 cells treated with canagliflozin (CANA) (10–30 μM), radiation (0–8 Gy), or the combination (CANA + RT), (**b**) and its normalization to β-actin, n = 3. Two-way ANOVA were used for comparisons between groups. **c** immunoblotting, quantification, and (**d**) cell proliferation assay with Roxadustat (Roxa) (5–30 μM, a hydroxylase inhibitor that stabilize HIF-1α), canagliflozin (30 μM), and the combination (Roxa/CANA (30 μM/30 μM)) to evaluate HIF-1α activity in PC3 cells, n = 3. Ordinary one-way ANOVA with the post hoc Tukey's multiple comparisons test was used for comparisons, asterisks represent of p value, *p < 0.05, **p < 0.01, ***p < 0.001, and ****p < 0.0001. The data show Mean ± SEM; n = 3.

biological processes (GOs) regulated by canagliflozin in irradiated cells included DNA repair, hallmark E2F targets, DNA replication, POLO-like kinase, G1-S specific transcription, hallmarks of G2-M checkpoint and Rho-Cdc42-Rac1 GTPase cycle for the downregulated pathways (Fig. 8c) (see Supplementary data file 2.xlsx for complete list). Translation elongation, response to starvation, aerobic respiration electron transport, hallmark oxidative stress, fatty acid metabolism, OxPhos and TCA cycle, apoptotic pathway in response to ER stress, and p53 pathway gene sets, are examples of the upregulated pathways. Heatmaps of the regulation of genes involved in cell cycle, DNA replication and p53 pathway are shown (Fig. 8d), cellular response to starvation is shown in (Fig. s4a). Importantly, canagliflozin specifically opposed radiotherapy action and induced a differential downregulation of selected genes such as *ROCK2, EP300, KLF11, GJA1, and TAOK3*, which are involved in tumor growth and survival (Fig. 8d). *ROCK2*, a

kinase activated downstream of the small GTP-binding protein RhoA, regulates actin cytoskeleton, cell polarity, and centrosome duplication while also activating Erk1/2 and a number of downstream kinases and transcription factors involved in proliferation, cell division, and survival[58]. *EP300* (E1A-associated cellular p300 transcriptional co-activator) is a histone acetyltransferase that regulates transcription via chromatin remodeling and is a co-activator of HIF-1α[59].

Overall, canagliflozin enhanced the ability of radiotherapy to regulate key genes involved in cell cycle, DNA replication and the p53 pathway in irradiated cells but, remarkably, opposed the activity of radiotherapy to stimulate genes involved in survival and radio-resistance, effects that likely contribute to an improved anti-tumor activity in irradiated PC tumors. This shows that canagliflozin mediates a transcriptional reprogramming that is clearly distinct from that induced by genotoxic stress and could address radio-resistance at the transcriptional level.

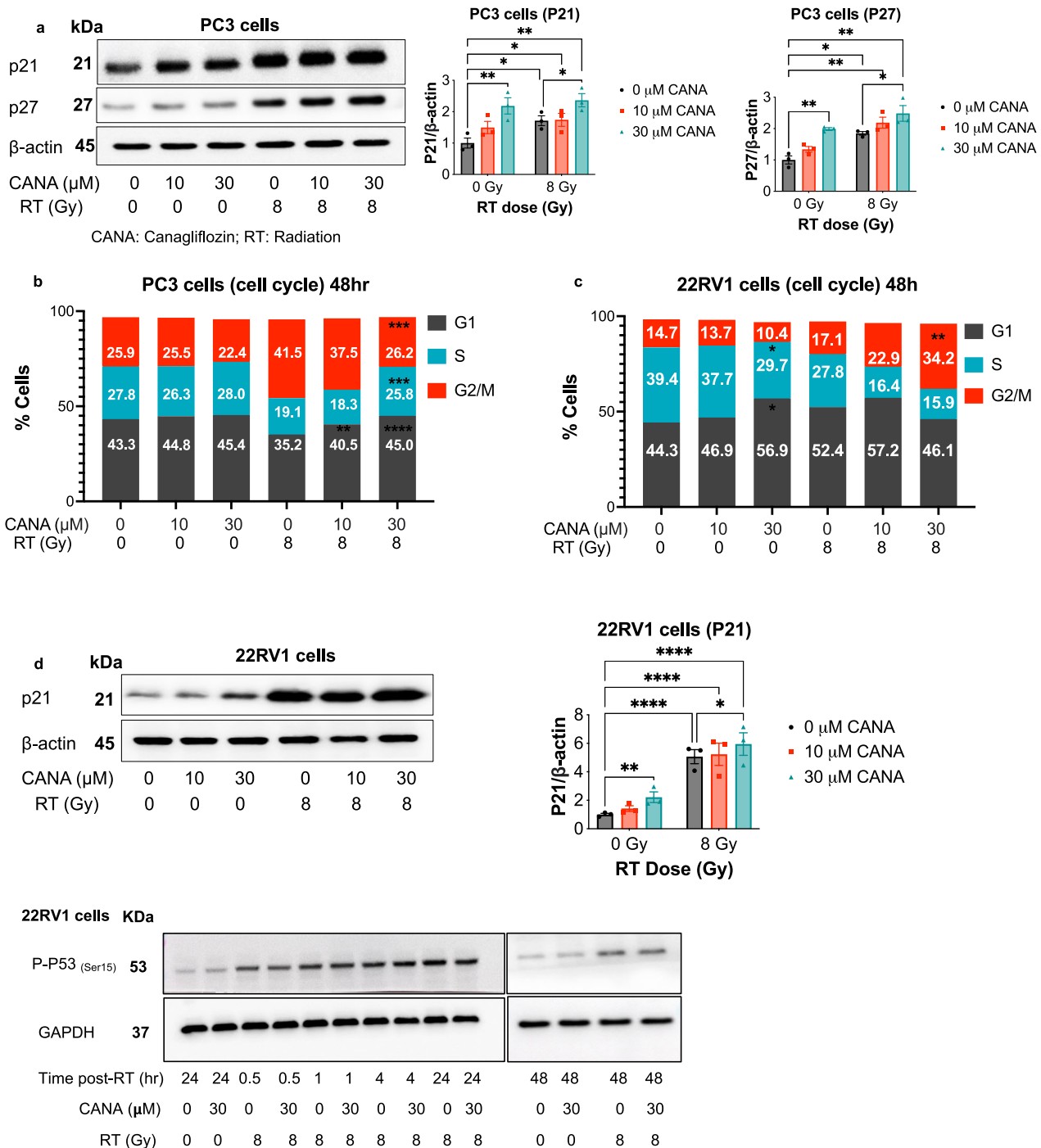

**Fig. 7 Cell cycle regulation. a** Immunoblotting and quantification of cell cycle checkpoints in PC3 cells. **b** Cell cycle analysis of PC3 cells, and (**c**) 22RV1 cells treated with canagliflozin (0–30 μM) and/or radiation (RT) (0–8 Gy). **d** Immunoblotting and quantification of cell cycle checkpoints in 22RV1 cells. Two-way ANOVA with the post hoc Tukey's multiple comparisons test was used for comparisons between groups. asterisks represent of $p$ value, *$p < 0.05$, **$p < 0.01$, ***$p < 0.001$, and ****$p < 0.0001$. Data are shown as Mean ± SEM; $n = 3$.

**Canagliflozin's transcriptional regulation points to involvement of ETS and E2F family transcription factors. Association with PC prognosis**. To better understand the underlying processes supporting canagliflozin's transcriptional program we subjected RNAseq datasets from non-irradiated PC3 and irradiated 22RV1 cells to transcription factor enrichment analysis, using the Cytoscape-*iRegulon* module[60]. Figure 9a–d illustrates key transcription factors regulated by canagliflozin and the breadth of genes and cellular processes found in our dataset to be regulated by the drug. The complete list of transcription factors

detected to be significantly regulated by canagliflozin is shown in Supplementary Table 1. Non-irradiated and irradiated PC cells respond to canagliflozin with induction of the ETS family transcription factor *ELF3*, a repressor of androgen receptor action[61], which drives expression of ribosomal proteins, OxPhos, cytokines, fatty acid oxidation enzymes, and the induction of phosphatase genes (Fig. 9a). However, in irradiated cells transcription factors regulated by *ELF3*, (such as *ATF3, ATF4, FOS, ETV4*) are also induced by canagliflozin, which, in concert with their effector gene network, appear to further propagate a transcriptional

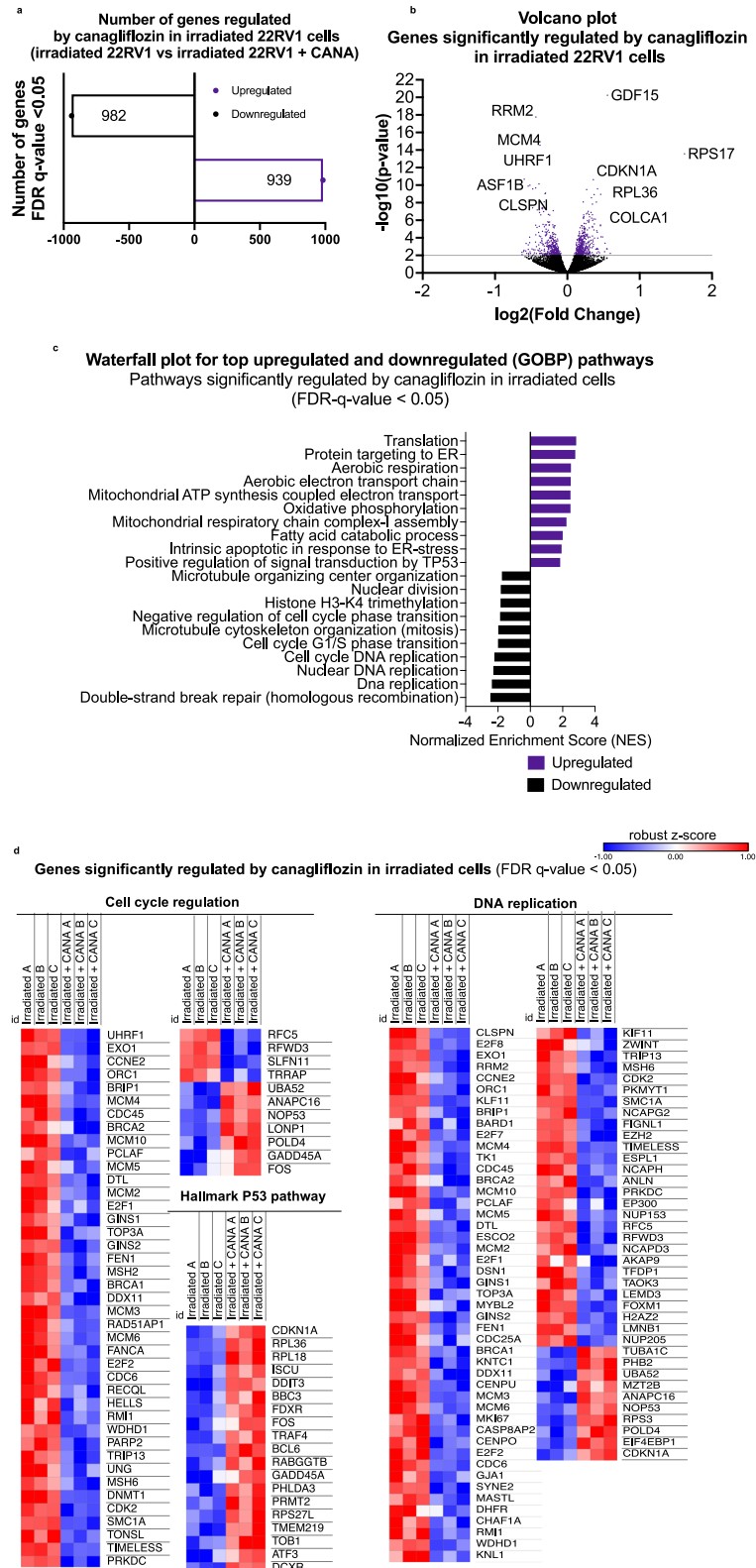

profile that includes ER-stress response and regulation of cell cycle and apoptosis (Fig. 9c). Further, canagliflozin suppresses another ETS domain transcription factor, *ELK1*, that supports androgen receptor signaling, and may be responsible for the drug's extensive transcriptional reprogramming involving repression of genes involved in mTOR, MAPK, hypoxia, and Wnt pathways (Fig. 9b). In addition, canagliflozin blocks expression of

transcription factors of *E2F* family and *GATA2*, known to regulate genes involved in cell cycle, DNA replication, mitosis, protein synthesis and survival[62] (Fig. 9b, d). For PC3 and 22RV1 RNAseq normalized count lists, please see Supplementary data file 3.xlsx and Supplementary data file 4.xlsx.

To begin validating the significance of canagliflozin's transcriptional program, we pursued an early analysis of the potential

**Fig. 8 Regulation of gene expression by canagliflozin (CANA) in irradiated 22RV1 cells.** RNAseq was performed 24 h after 22RV1 delivering radiation (5 Gy) with or without CANA (10 $\mu$M). In the supplementary data, the 22RV1 cells were treated with DMSO or CANA as control groups. **a** Number of significantly ($p < 0.05$) upregulated and downregulated genes by canagliflozin (10 $\mu$M) in irradiated cells (**b**) A volcano plot depicts the total number of genes that are up- and down-regulated by canagliflozin in irradiated cells subjected to RNAseq analysis. The top modulated genes are indicated. Significantly modulated genes are represented by purple dots, whereas genes with $p$-values greater than 2 ($-\log10(p\text{-value})$) are separated by a gray line and have a False Discovery Rate (FDR) $q$-value < 0.05. A negative log2 fold change indicates that genes are downregulated, whereas a positive log2 fold change indicates that genes are upregulated. **c** Waterfall plot depicts the top 10 significantly downregulated and upregulated Gene Ontology Biological Process (GOBP) pathways by canagliflozin in irradiated cells, FDR $q$-value < 0.05. **d** Heatmap illustrations provide a detailed description of gene regulation in the cell cycle, DNA replication and hallmark of p53 pathways, after 24 h of treatment, analyzed in triplicates (A, B, C), FDR $q$-value < 0.05. Normalized log2 feature counts are used to represent gene expression (with robust z-score). The blue to red scale represents downregulated to upregulated from −1 to 1. The volcano plot, waterfall plot, and heatmaps in Fig. 8 provide a visual representation graphs of the comparison between irradiated 22RV1 cells with (5 Gy) that were treated with canagliflozin (10 $\mu$M) and non-treated with canagliflozin.

prognostic value of genes regulated by canagliflozin in PC clinical trial datasets using the ProgGeneV2 and PCTA engines. Genes downregulated significantly by canagliflozin treatment in PC3 or 22RV1 cells (*TRIM28, RRM2* and *MCM4*: Figs. 3b, 8b) are associated with poor overall survival when expressed at levels higher than the median, in the Swedish GSE16560 cohort (see HRs and *P* values in Fig. 9e). Interestingly, the combined presence of increased levels of transcript of all three genes had higher HR (2.57) for worse overall survival ($P = 0$) compred to each gene alone. We found similar association with overall survival for *RRM2* mRNA levels in the TCGA dataset (FDRq=0.0126) (Fig. s5a). Higher *RRM2* expression is associated with worse biochemical recurrence-free-survival in the GSE70769 cohort (Fig. s5b). Further, transcription factors downregulated by canagliflozin, *E2F1* and *E2F3*, are associated with poor overall survival and, similarly, the combined signature of the two shows greater association with poor outcomes (Fig. 9e).

The Swedish Watchful Waiting cohort (GSE16560)[63] is one of the best repositories of prognostic information for PC patients, containing up to 30 years of follow-up data. The observation that low levels of the effector genes and transcription factors suppressed by canagliflozin are associated with improved survival, and that combined signatures of those genes do so with greater statistical significance (Fig. 9 and Table s1a) strengthens the notion that this agent has increased potential to provide clinical efficacy. This is further strengthened by the finding that canagliflozin-regulated genes are associated with survival outcomes in TCGA and GSE16560 PC datasets also.

Overall, our data points to ETS and E2F family transcription factors as potential mediators of canagliflozin's transcriptional program, which regulates favorably genes associated with PC prognosis.

**Conclusion.** This study provides substantial new evidence that the SGLT2 inhibitor canagliflozin, a widely used diabetes therapy, provides, within its therapeutic window, tumor suppressive and radio-sensitizing activity in PC through a complex multi-target mechanism. Canagliflozin suppresses survival and tumor growth in both androgen-sensitive and—insensitive PC models and demonstrates increased activity in radio-resistant PC cells compared to parental controls. It suppresses mitochondria respiration, but unlike other OxPhos inhibitors, it does not alter extracellular acidification. This is achieved, at least in part, through induction of mitochondria complex-I blockade and suppression of HIF-1α (see model Fig. 10). In both non-irradiated and irradiated cells, canagliflozin mediates an extensive reprogramming of PC cell transcriptional activity pointing to the involvement of ETS- and E2F-family transcription factors.

In recent years, clinical trials investigated the safety and efficacy of novel therapeutics alone and in combination with radiotherapy to control PC progression. Modern anti-androgen

therapy trials demonstrated improvements in survival in combination with androgen-deprivation or chemotherapy and have changed practice in metastatic PC[64–66]. Immune checkpoint inhibitor therapies were also examined with limited success[67,68]. Ongoing studies examine these therapies in combination with radiotherapy[69–71].

Given that canagliflozin is an approved medication, well-tolerated by non-diabetics, and without known interaction with cancer therapeutics, the data presented here provide a valid basis for clinical investigation of canagliflozin in prostate cancer prevention and definitive therapy in combination with radiotherapy. Demonstrating clinical benefit in early phase trials could open opportunities for further investigation of canagliflozin in combination with anti-androgen and immune check-point inhibitor therapies.

**Materials.** All standard chemicals were purchased from Fisher Scientific (Toronto, ON), Sigma Aldrich (Oakville, ON), Bio-Rad (Mississauga, ON), and Bioshop (Burlington, ON).

For tissue culture assays, Canagliflozin ($C_{24}H_{25}FO_5S$) was purchased from MedChemExpress (MCE) (Monmouth Junction, NJ). For in vivo studies, animals received clinical grade canagliflozin (Invokana 300 mg tablets, Jansen Inc.) incorporated into chow diet generated by Envigo (Indianapolis, IN). IACS-010759 ($C_{25}H_{25}F_3N_6O_4S$), BAY87-2243 ($C_{26}H_{26}F_3N_7O_2$), and Roxadustat ($C_{19}H_{16}N_2O_5$) were purchased from Cayman chemical (Mississauga, ON).

**Antibodies.** Primary and secondary antibodies used in immunoblotting were purchased from Cell Signaling Technology (Whitby, ON) (Supplementary Table 2).

**Experimental models**
*Cells.* PC3, 22RV1 and DU145 cells were purchased from ATCC. DU145-RR cells were generated by serially treating wild-type DU145 cells with 2 Gy daily fractions of radiotherapy (Monday-Friday) to a total of 118 Gy, as previously described[33]. LnCap cells were provided by Dr. Damu Tang, (McMaster University).

*Animal.* Athymic BALB/C nude mice (NU(NCr)-*Foxn1^nu^*, lacking T-lymphocytes) were purchased from Charles River (Wilmington, MA). NRG mice (NOD.Cg-*Rag1^tm1Mom^Il2rg^tm1Wjl^*/SzJ, lacking T- and B-lymphocytes, and NK cells) were purchased from Jax Laboratories (Bar Harbor, ME) and bred in-house.

**Methods**
**Cell Culture.** Cells were authenticated using short tandem repeat DNA profiling, and the amplified DNA sequences were compared to the reference cell database, with a match of more than 80% being acceptable. Following that, cells will be checked for

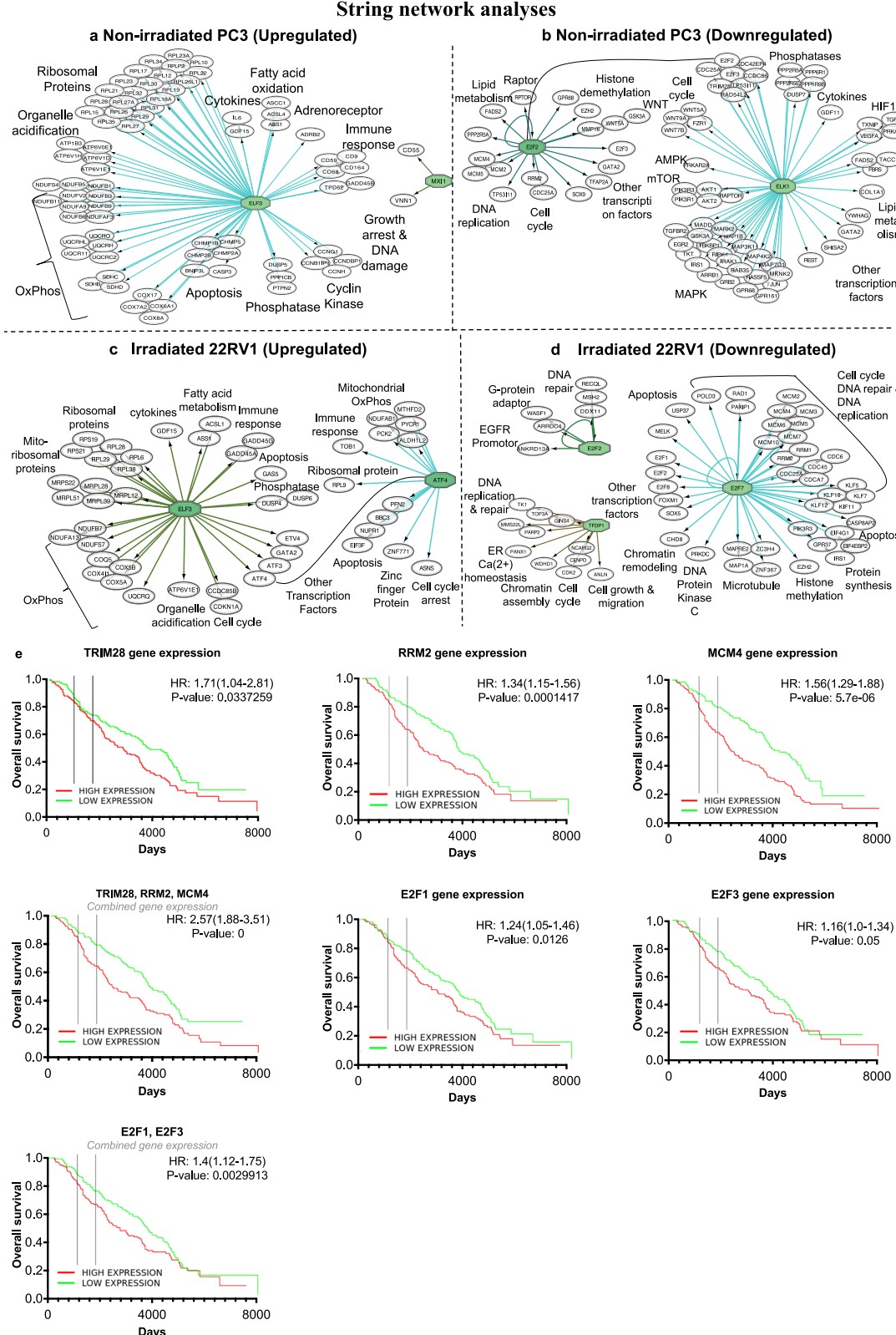

**String network analyses**

a Non-irradiated PC3 (Upregulated)  b Non-irradiated PC3 (Downregulated)

c Irradiated 22RV1 (Upregulated)  d Irradiated 22RV1 (Downregulated)

e TRIM28 gene expression — HR: 1.71(1.04-2.81) P-value: 0.0337259

RRM2 gene expression — HR: 1.34(1.15-1.56) P-value: 0.0001417

MCM4 gene expression — HR: 1.56(1.29-1.88) P-value: 5.7e-06

TRIM28, RRM2, MCM4 Combined gene expression — HR: 2.57(1.88-3.51) P-value: 0

E2F1 gene expression — HR: 1.24(1.05-1.46) P-value: 0.0126

E2F3 gene expression — HR: 1.16(1.0-1.34) P-value: 0.05

E2F1, E2F3 Combined gene expression — HR: 1.4(1.12-1.75) P-value: 0.0029913

mycoplasma negative contamination (Lonza), and all subsequent studies will have a passage number of no more than 20. PC3 cells were cultured in RPMI and 22RV1 and LnCap in ATCC-modified RPMI media supplemented with 10% FBS and 1% antibiotic-anti-mycotic. DU145 and DU45-RR cells were cultured in DMEM media supplemented with 10% FBS and 1% antibiotic-anti-mycotic.

**Proliferation and clonogenic survival assays**. For proliferation assays, 96-well plates were seeded at 500 cells/well. For clonogenic assays, 12-well plates were seeded 500–4000 cells/well. Cells were incubated overnight, followed by canagliflozin treatment (0–30 $\mu$M) and radiotherapy (0–8 Gy) 6 h h after pre-incubation without or with canagliflozin. For proliferation assays, cells were incubated for approximately 5 days or until control wells reached

**Fig. 9 Prognostic analysis. a–d** String network analyses for transcription factor (green color) interaction with target genes and other transcription factors (white color) generated from RNAseq data of (**a, b**) non-irradiated PC3 and (**c, d**) irradiated 22RV1 cells. **a** Upregulated, and (**b**) Downregulated transcription factors and their target genes in PC3 cells treated with (10 μM) canagliflozin (CANA vs control) for 24 h. Genes from PC3 RNAseq where filter with False Discovery Rate (FDR) *q*-value < 0.05. **c** Upregulated, and (**d**) Downregulated transcription factors and their target genes in 22RV1 cells (comparison between irradiated 22RV1 cells with (5 Gy) that were treated with canagliflozin (10 μM) vs non treated irradiated cells). All significant genes filtered with FDR *q*-value < 0.05 were extracted from RNAseq data and entered into the iRegulon program, which predicts the most important transcription factors that regulate the majority of these upregulated or downregulated significant genes. **e** Kaplan–Meier survival curves validating the association of clinical outcome of the top downregulated genes (*TRIM28, RRM2, MCM4*) and transcription factors (*E2F1, E2F3*) expression by canagliflozin in prostate cancer patients from GSE16560. Cohort divided at the median of gene expression. The downregulated genes were selected from our RNAseq analysis of non-irradiated PC3 and irradiated 22RV1 cells.

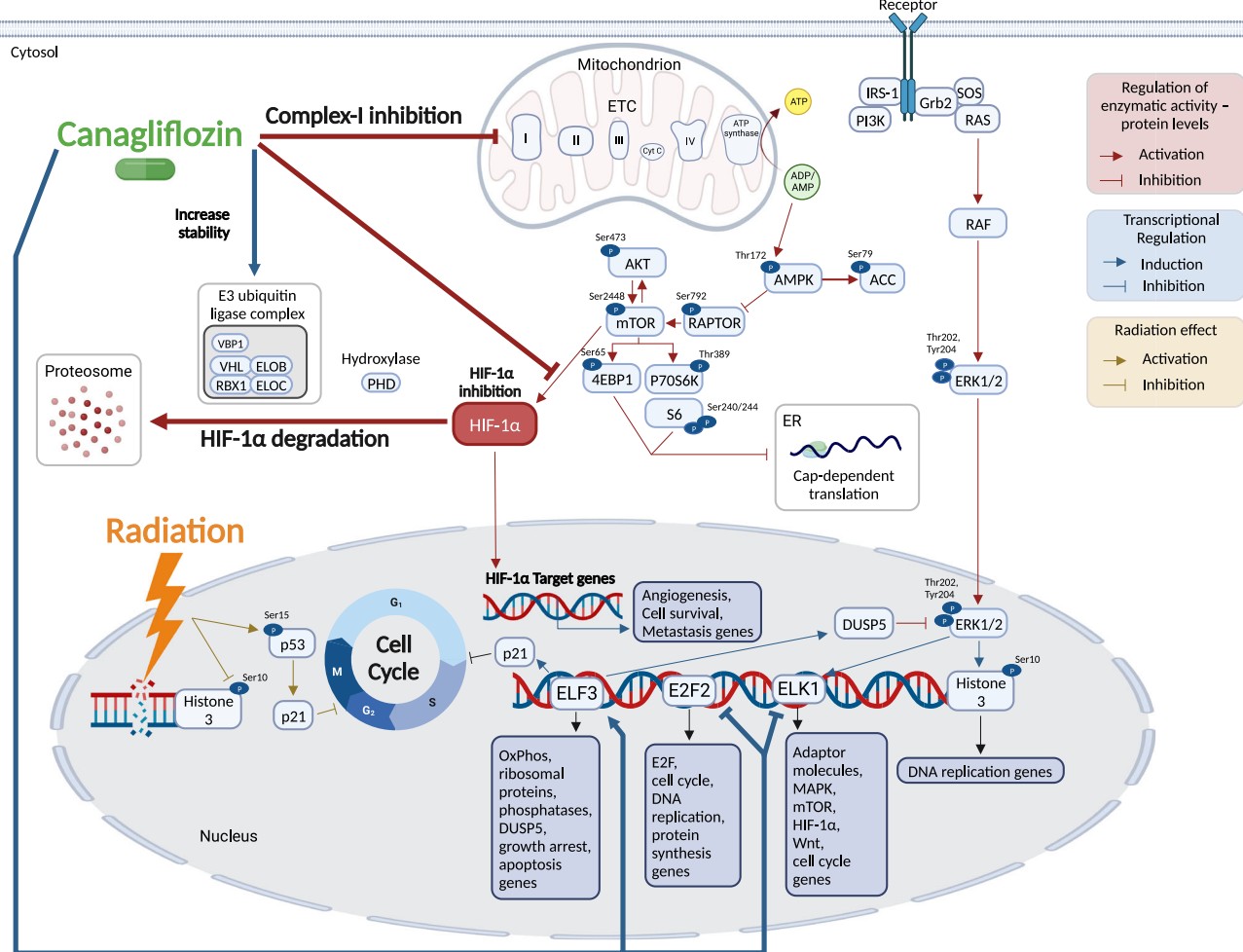

**Fig. 10 Cellular pathways modulated by Canagliflozin in prostate cancer cells.** The work presented here suggests that canagliflozin mediates its anti-tumor activity through a complex multi-target pathway. Canagliflozin inhibits mitochondria respiration through complex-I, leading to activation of the metabolic stress sensor AMPK. This leads to suppression of ACC-de novo lipogenesis pathway (by increasing the P-ACC(ser[79]), the inactive form of ACC), Akt/mTOR/p70S6K – 4EBP1—protein synthesis axis and reduction of HIF-1α levels. Complex-I inhibition and suppression of HIF-1α levels contribute to at least part of canagliflozin's anti-tumor activity. Canagliflozin upregulates the E3-ubiquitine p-VHL complex genes, which may contribute to enhanced HIF-1α degradation. On the other hand, canagliflozin de-activates Erk signaling, likely in part through induction of the nuclear and dual-phosphatase gene (DUSP5). Consistently, the drug blocks, phosphorylation of Histone3 p-H3(ser[10]), which controls DNA replication and induction of cell cycle checkpoints. Canagliflozin mediates a marked re-programming of prostate cancer transcriptional activity likely through regulation of ELF3, E2F, and ELK transcription factors known to regulate OxPhos, phosphatase, apoptosis, fatty acid metabolism, hypoxia and Wnt pathway genes. Overall, canagliflozin suppresses biosynthetic, survival and radio-resistance pathways at the transcriptional and post-translational level and mediates tumor suppression and improvement of response to radiotherapy.

80% confluency. Cells were then fixed (10% formalin), stained with crystal violet, dried and subjected to absorbance was reading at 762 nm using SpectraMax iD5 system (Molecular Devices, San Jose, California).

For clonogenic assays, cells were incubated for approximately 7 days or until colonies of at least 50 cells were visible in untreated wells. These cells were fixed, stained with crystal violet and colonies were then counted.

To determine the mode of interaction for therapy combinations (additivity, synergism, or antagonism) we used SynergyFinder (https://synergyfinder.org) with the Highest Single Agent mathematical modeling (HSA)[34]. The HSA model assumes synergy if the effect of a combination is greater than that of any of the single drugs alone. HSA mean score of 10 or higher indicates synergism, a score between (+10 and −10) indicates additivity, and a score of less than −10 indicates antagonism. RAD-ADAPT software was used for modeling clonogenic assay data in RT biology, using the linear-quadratic approach[72].

**Animal studies**. All experiments were approved by the McMaster Animal Ethics Committee and conducted following the guidelines of animal research (AUP # 16-12-41 and 20-12-47). The right flanks of 6–8-week-old male mice were injected with $1 \times 10^6$ PC3 or 22RV1 cells. Two distinct approaches were taken in these experiments. PC3 xenografts were generated in athymic BALB/C nude mice and tumor growth kinetics were studied with an experimental endpoint set for all animals when control animal tumors reached an average tumor size of 1250 mm³. Tumors were extracted from all animals 1-2 days after the endpoint was reached. 22RV1 tumors were grafted into Jax Labs NRG mice. This experiment was designed with endpoints set individually for each animal when the tumor reached 2200-2500 mm³ (tumors were extracted after xenograft reached the endpoint).

Canagliflozin was provided using clinical-grade drug (Invokana 300 mg tablets, Jansen Inc.) incorporated into Chow diet at 416.7 ppm (purity 83.3%: 347.24 mg of canagliflozin per kg diet). Based on animal body weights ranging 22-30 g over the experimental period and the calculated daily intake of chow diet of 2.5-6 g/day (estimated 60% intake vs 40% waste), the canagliflozin diet was calculated to deliver 40–70 mg/kg animal body weight per day.

Mice were randomly assigned to no-treatment (Control), radiation (RT) (5 Gy), canagliflozin diet (CANA) or canagliflozin and radiation (CANA + RT). Mice in the control group did not receive any treatment, mice in the radiation group received 5 Gy of radiation once tumors reached 50 mm³ for BALB/C nude mice or 200 mm³ for NRG mice. In the canagliflozin group mice were given a canagliflozin diet (416.7 ppm) and mice in the combined treatment were put on a canagliflozin diet for 3 days before receiving radiation. In the PC3 BALB/C xenograft module, once the control group tumors volume reached 1000 mm³, all mice were euthanized, and tumors were extracted and a half was fixed in 10% formalin for 48 hours then transferred to 70% ethanol solution, for further analysis. The other half of the tumor was frozen under −80 °C, for further analysis. In the 22RV1 NRG xenograft module, mice were euthanized only when the tumor volume reached 2500 mm³ and the survival curve was plotted. Tumor mass volume in mm³ was monitored throughout the study using a calliper and was measured using the following formula: $L/2 \times W2$.

**Immunoblotting**. Cells seeded in 6-well plates were treated with the indicated doses of canagliflozin and/or radiotherapy (RT). For combined treatments, canagliflozin treatments were initiated 4 h before radiotherapy delivery, and incubation was stopped 48 h later. Then, cells were lysed using a lysis buffer containing 50 mM HEPES, 150 mM NaCl, 100 mM NaF, 10 mM Na-pyrophosphate, 5 mM EDTA and 250 mM of sucrose, 1 mM Dithiothreitol (DTT), 1% Triton-X, 1 mM Na-orthovanadate and 1% complete protease inhibitor. Protein concentration was then measured using a BCA protein assay protocol. Samples were then diluted with 4x SDS sample buffer containing 40% glycerol, 240 mM Tris-HCl pH 6.8, 8% SDS, 0.04% bromophenol blue, and 5% β-mercaptoethanol to a final concentration of 1 $\mu$g/$\mu$L. Lysates were then run-on polyacrylamide gels for protein separation to occur. After separation, samples were being transferred on a PVDF membrane then the membranes were blocked with 5% BSA for 1 h at room temperature. Membranes were then incubated with primary antibodies overnight. The following day, membranes were washed with TBST, then incubated with secondary antibodies for 1 h and afterthat membranes were then washed with TBST. Finaly, blots we analyzed with Clarity™ Western Enhanced Chemi-Luminescence (ECL) Substrate (Bio-rad, Mississauga, ON) and SuperSignal™ (Thermo Scientific, Mississauga, ON) using a Vilber Fusion-FX imager (Marne-la-Vallée cedex 3, France), and then images were processed using ImageJ software (Version 1.53t) for quantification. Density values for each marker were normalized to β-actin or GAPDH.

**Mitochondrial respiration assay**. The oxygen consumption rate (OCR) and extracellular acidification rate (ECAR) were measured in PC3 cells seeded at 20,000 cells per well, treated with the indicated canagliflozin and radiotherapy doses, and analyzed 48 hours later using the Agilent Technologies Seahorse XFe96 extracellular flux analyzer system (Santa Clara, CA). Briefly, incubation medium was changed one hour before the start of the assay to Seahorse XF medium, which was supplemented with 25 mM glucose, 2 mM glutamine, and 1 mM sodium pyruvate. Different ETC complex inhibitors (oligomycin, FCCP, and rotenone/antimycin-A) were then sequentially added during the assay at pre-optimized concentrations of 1.5 $\mu$M, 1 $\mu$M, and 0.5 $\mu$M, respectively. Then, OCR, ECAR, basal respiration, maximal respiration, ATP production, non-mitochondrial respiration, spare respiratory capacity, OCR:ECAR ratio were calculated[73].

The following formulas were used to calculate mitochondrial function variables (basal respiration, maximum respiration, mito-ATP production, and basal ECAR:OCR); Basal respiration= final rate measurement before the initial injection–non-mitochondrial respiration rate. Maximal respiration= maximum rate measurement after FCCP injection–non-mitochondrial respiration. ATP production= final rate measurement before Olig. injection–minimum rate measurement after Olig. injection. The following formula was used to convert ATP production to mitochondrial ATP production rate: MitoATP production rate (pmol/ATP/min)= $OCR_{ATP}$ (PMOL/$O_2$/min) x 2 (pmol O/pmol $O_2$) x P/O (pmol ATP/pmol O), where P/O was adjusted to a verified value of (2.75). Basal OCR:ECAR ratio=The OCR mean from the last three baseline data points, divided by the ECAR mean of the same last three baseline data points. The default buffer factor of 2.4 was taken into account and implemented prior to performing the calculations.

**Glycolytic rate assay**. For the Glycolytic Rate Assay in PC3 cells, the medium was switched to Seahorse-XF RPMI medium supplemented with 5 mM glucose, 2 mM glutamine, and 1 mM sodium pyruvate. The Seahorse XFe96 analyzer was used to record basal ECAR, and post-injection of 0.5 $\mu$M rotenone/antimycin-A and 50 mM 2-deoxyglucose (2-DG) (XF Cell Mito Stress Test Kit), respectively. Basal proton efflux rate (PER) was calculated using Wave desktop software, using data from the Seahorse glycolytic rate assay. The default buffer factor = (2.4) was taken into consideration and was applied before calculations. To normalize of OCR/ECAR/ER values obtained from each assay to cellular content, after the mitochondrial respiration or glycolytic rate assays, cells were fixed with 10% formalin, stained with 0.5% crystal violet, dried, and measured for absorbance at 762 nm using the SpectraMax iD5 system (Molecular Devices, San Jose, California) to determine DNA content. The obtained absorbance values, for each well, were then used to normalize the OCR/ECAR/PER values from the same well.

**Cell cycle analysis**. PC3 and 22RV1 cells were seeded in a 10 cm dishes ($0.5 \times 10^6 – 2 \times 10^6$/dish), incubated overnight and treated without or with canagliflozin (CANA) (0–30 μM) and or radiotherapy (RT) (0–8 Gy). After treatments, cells were incubated for 48 h until they were 50–60% confluent. Cells were then harvested and washed with cold PBS buffer, fixed in 70% ethanol, and stored at −20 °C. Before analysis cells were centrifuged, EtOH was aspirated, cells were washed with PBS and stained with propidium iodide (ThermoFisher FxCycle PI) used to stain cells for 30 minutes followed by flow cytometry analysis, using a Cytoflex LX flow cytometer (Beckman Coulter, Mississauga, ON) (Core Flow Facility, McMaster University). For the gating strategy, we utilized forward scatter and side scatter to identify viable, single-cell events. This cell population gate was positioned on PI-area vs PI-height to eliminate doublets from the analysis. Data analysis was performed using FlowJo software (Version 10.8.0, FlowJo LLC, Ashland, OR).

**RNA sequencing analysis**. PC3 or 22RV1 cells were seeded at $3 \times 10^5 – 5 \times 10^5$ cells/well in 6-well plates and treated with 10 μM canagliflozin, 5 Gy radiation or both. 24 hours later cells were lysed in Trizol RNA isolation reagent (200 μl per well, Thermo Fisher Scientific, CA). RNA was extracted using chloroform (200 μl), centrifugation at $12,000 \times g$ (15 min at 4 °C) and precipitation using isopropanol. RNA pellets were rinsed with 70% ethanol and resuspended with 40 μl RNase-free water. cDNA was prepared using ReverAid™ First Strand cDNA Synthesis Kit (Thermofisher, CA) and stored at −20 °C. Then Illumina HiSeq 1500 (Illumina; San Diego, CA, USA), next-generation sequencing was performed at the McMaster Genomics Facility (Farncombe Institute McMaster University) with HiSeq Rapid v2 flow cell to eliminate lane-specific effects. Single-end reads were performed at 1x50bp configuration with the goal of generating an average of 25 million reads per sample.

The use-galaxy platform (https://usegalaxy.org/) was used for analysis. Raw sequencing data were assessed for quality (such as GC content, PHRED scores, synthetic aptamer content, and sequence length) using FastQC. (https://www.bioinformatics.babraham.ac.uk/projects/fastqc/). Low-quality reads were trimmed using the trimmomatic tool Cutadapt to improve the quality of sequences via trimming and filtering[74], with a Minimum length and Quality cut-off set to "20", Minimum overlap length set to "3", and 5′ adapter sequence "AGATCG-GAAGAGCACACGTCTGAACTCCAGTCA". Then the final set of reads is aligned to the human gene (hg38) and specified strand information set to "reverse (R)" using the HISAT2 tool[75]. The number of reads mapping to each gene was identified using feature Counts[76], with stranded set to "reverse", and feature type filter to "exon". DESeq2 tool was used to determine differentially expressed features from count tables and normalized these data[77]. The human gene annotation reference file was obtained from Ensembl gene using the annotateMyIDs tool. A filter tool was used to filter the adjusted $p$-value (FDR; false discovery rate) and set to "<= 0.05", then the r-log normalized list was generated.

A rank list was created from the normalized gene list using the equation "negative log10 of the $p$-value". The ranked list was used as input for Gene Set Enrichment Analysis (GSEA)[78] at "https://www.gsea-msigdb.org/", using 1000 permutations, a false discovery rate cut-off of 5%, and a human gene set database selected "Human_GOBP_AllPathways_no_GO_iea_October_06_2021_-symbol.gmt" from the Bader Lab (UOT) at "https://download.baderlab.org/EM_Genesets/October_06_2021/Human/symbol/". GSEA output was conducted using Cytoscape software[79]. Heat maps were generated using Morpheus software ("https://software.broadinstitute.org/morpheus").

**Transcription factor (TF)-target gene regulatory networks**. We used a TF prediction tool to detect key transcription factors involved in canagliflozin activity to better understand the underlying mechanism of action. Putative TFs relevant to our RNAseq data were identified using the Cytoscape-iRegulon plug-in[60]. Significantly upregulated or downregulated genes were imported (HGNC symbol, FDR $q$-value (FDRq) less than 0.05, and fold change more than 1). The iRegulon relevant parameters used were: enrichment score threshold = 3.0, ROC threshold for AUC calculation = 0.03, rank threshold = 5000, minimum identity between orthologous genes = 0.0 and maximum FDR on motif similarity = 0.001, and ranking parameters were left at their default values. To improve prediction accuracy and obtain better representation of the changes observed in RNAseq data, we only selected TFs with FDRq<0.05. String networks with representative TF-target genes were generated using the STRING plug-in Cytoscape[80].

**Prognostic analysis**. The ProgGeneV2 (http://www.progtools.net/gene/index.php) and Prostate Cancer Transcriptome Atlas (PCTA) (http://www.thepcta.org) engines were used to analyze open-source mRNA expression and survival data from the Swedish-Watchful-Waiting cohort[81] and GSE16560[63] vs The Cancer Genome Atlas (TCGA)[82] and Cancer Research UK Cambridge Institute cohort (GSE70769)[83], respectively. The GSE16560 dataset was analyzed with ProGeneV2. It contains 281 cases from the Swedish-Watchful Waiting cohort. Men in the GSE16560 cohort were diagnosed with localized PC at clinical stage T1-T2, Mx, N0. Overall survival in this cohort was determined by bifurcate gene expression at the median. PCTA offers extensive transcriptome data from 1321 clinical specimens from 38 PC cohorts[84]. The Cancer Genome Atlas (TCGA) and the Cancer Research UK Cambridge Institute cohort (GSE70769) were chosen and analyzed using the PCTA webtool for Biochemical Recurrence Free-Time Analysis (BCA).

**Immunohistochemistry (IHC)**. Fresh tumor tissues from 22RV1 or PC3 xenografts were fixed in 10% neutral buffered formalin for 48 hours before being transferred to and stored in 70% ethanol. The tissues were processed and embedded in paraffin blocks by the Research Histology Core Laboratory (McMaster University) according to standard protocols. Tissue blocks were sectioned at a thickness of 5 μm. Tissue sections were deparaffinized and rehydrated in xylene and ethanol, followed by endogenous peroxidase blocking and heat antigen retrieval in citrate buffer with pH=6 (Sigma-Aldrich#C9999). Tissues were blocked in 10% normal goat serum (Vector laboratories#S-1000-20) and incubated overnight at 4 °C with either non-specific (negative control) serum or anti-P-H3(Ser[10]), anti-P-ACC(Ser[79]), anti-P-CC3(Asp[175]) rabbit antibodies (cell signaling #9701, #11818, #9661) with 1:200, 1:700, 1:400 dilution, respectively. Then, tissues were incubated with biotinylated goat-anti-rabbit IgG secondary antibody (1:500 dilution) (Vector laboratories#BA-1000) and streptavidin peroxidase (1:50 dilution) (Vector laboratories#SA-5004). Antigen-antibody complex was detected using the Nova Red kit (Vector laboratories#SK-4800). Hematoxylin (Abcam#245880) was used as counterstaining. For evaluation of P-H3(Ser[10]), 20 random High-power field (HPFs) (40x) of each slide (each mouse) were quantified by using Olympus BX-40-F4 microscope (Breinigsville, PA) and the average percentage of positive nuclei per slide was taken by the researcher. H-scores were calculated for all slides as the product of cell percentage and staining intensity, yielding an ordinal value with 300 possible values. For evaluation of CC3 and necrosis ratio, 10 random HPFs (40x) of each slide were digitally quantified using ImageJ software[85]. To quantify the necrosis % in the 22RV1 xenograft

tumor, all slides were stained with H&E and the whole section was quantified using ImageJ software, following the ImageJ user guide for tissue quantification found on the NIH ImageJ website (https://imagej.nih.gov/ij/index.html). We calculated necrotic area as follows: Necrotic Area = Total Area—Viable Tissue Area, and Necrotic Tissue Percentage = Necrotic Area/Total Area.

*Statistics and reproducibility.* The values are reported as means, with bars indicating the standard error of means ( ± SEM), as well as the results were analyzed for significance using GraphPad Prism software (Version 9, San Diego, CA). At least three independent experiments were performed ($n \geq 3$) to ensure robustness and reproducibility. Two-way ANOVA with the post hoc Tukey's multiple comparisons test was used to analyze the results, unless otherwise specified. A repeated-measures ANOVA was used for all bodyweight, food intake, water intake plots and IHC data. In some cases, as indicated in the figure legends, the Student's $t$ test or one-way ANOVA were used. Significance was accepted at $p \leq 0.05$, where $*p < 0.05$, $**p < 0.01$, $***p < 0.001$ and $****p < 0.0001$. The "IC$_{50}$" values for the drug's inhibitory effect were calculated using a non-linear regression model.

**Reporting summary.** Further information on research design is available in the Nature Portfolio Reporting Summary linked to this article.

## Data availability

The data supporting the findings of this study can be found in the paper and its supplementary files (Supplementary figures and Tables.pdf). The original blots used for Figs. 4, 5, 6, and 7 can be found in Supplementary Fig. 6, which is included in the attached (Supplementary figures and Tables.pdf). Numerical source data for graphs/charts can be obtained from (Supplementary numerical source data.xlsx). All other data are available from the corresponding author on reasonable request.

The RNA-seq data, including both raw data and normalized data, has been submitted to the Gene Expression Omnibus (GEO) under the accession number GSE:239688. The RNA-seq data GSEA analysis and normalized feature count can be found in the paper as supplementary excel files as follows:

1. PC3 cells RNAseq (canagliflozin vs control) pathway analysis: Supplementary data file 1.xlsx.

2. 22RV1 cells RNAseq (canagliflozin and radiation) pathway analysis: Supplementary data file 2.xlsx.

3. PC3 cells normalized Count RNAseq List (Canagliflozin vs Control): Supplementary data file 3.xlsx.

4. 22RV1 cells normalized Count RNAseq List, (Control, RT(5 G), Canagliflozin(10uM), and Combination): Supplementary data file 4.xlsx.

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

## Acknowledgements
The authors would like to thank the Juravinski Cancer Center's Radiotherapy Program team for their assistance with cell and animal irradiation. This study is supported by funds from Hamilton Health Science Foundation, the Canadian Institutes of Health Research, and the Canadian Association of Radiation Oncologists.

## Author contributions
A.A.: Investigation, Formal analysis, Methodology, Data curation, Writing—original draft, review & editing. B.M.: Investigation, Formal analysis, Methodology. O.D.B.: Investigation, Methodology, review & editing. E.E.T.: Investigation, Methodology, review & editing. E.A.: Investigation, Methodology, review and editing. J.W.: Methodology. S.W.: Investigation, review and editing. K.S.: Methodology. G.M.: Methodology. T.F.: Methodology. A.M.: Methodology. S.K.L.: Methodology. T.B.: Methodology, review and editing. J.L.B.: Methodology—review and editing. G.R.S.: Methodology, Funding acquisition,

review and editing. T.T.: Conceptualization, Methodology, Supervision, Project administration, Writing—review and editing, Funding acquisition.

## Competing interests

The authors declare no competing interests.
