## [Peer Review File · Communications Biology]

Reviewers' comments:

Reviewer #1 (Remarks to the Author):

The paper by Ali and colleagues uncovers a role for the anti-diabetic drug canagliflozin in promoting sensitivity to radiotherapy in prostate cancer. The paper presents data that will stimulate studies on anti-cancer combination therapies incorporating canagliflozin. While the paper is well written in general, the manuscript could be improved by presenting a more integrated and cohesive story. Also, the manuscript would be improved by discussing the results and incorporating more recent literature on prostate cancer and canagliflozin. Please find below specific comments.

The mechanism of action presented for canagliflozin is that it inhibits complex I. Yet, combining canagliflozin with two other complex I inhibitors augments its anti-proliferative properties. It would be important to explore whether canagliflozin could inhibit other complexes in the electron transport chain. For example, one could examine whether combining canagliflozin with complex II inhibitors (malonate) or complex III inhibitors (myxothiazol/Antimycin A) augment its anti-proliferative properties. It would also be interesting to examine whether the anti-proliferative effects of canagliflozin are augmented by treatment with the anti-diabetic drug metformin that also inhibits complex I.

It is strange that ECAR does not increase upon OCR inhibition. This is a central node of bioenergetics regulation in cells. In order to assess the impact of canagliflozin on cellular energy stress, the levels of ATP, ADP and AMP should be quantified. The authors should also specify the buffering correction that was used for the Seahorse experiments.

Data are presented to show that canagliflozin inhibits mTOR signalling. The impact of combining mTOR inhibitors with canagliflozin on the proliferative potential of prostate cancer cells should be explored.

The use of part of the RNA seq data throughout the paper, while the data are actually discussed in the later parts of the paper, is confusing.

Line 66: The authors should consider using more recent citations.

Line 68-69: The authors should consider reframing this. More recent theories have been laid out.

Line 82: The link is not clear. The authors should introduce HIF and mTOR links with mitochondrial metabolism.

Line 89 and throughout: The authors should normalize phosphorylation annotation throughout the paper.

Line 117: The authors should consider calculating the actual IC50 using a non-linear regression model.

Line 236: Are OXPHOS measures in Fig4a corrected for cell content?

Line 242: OCR/ECAR ratio is a measure of metabolic phenotype, not of metabolic stress.

Line 529: These "cells" were fixed...

Line 579-595: Please specify how Seahorse data were normalized.

Lines 599-601: It looks like some part of the text was cut out.

Lines 616-618: Please specify the average amount of reads per sample, and if the analysis was done

using single- or pair-end sequencing. Furthermore, raw sequencing data should be made available through public repositories.

Figures

Overall, the graphs are very small and colors hard to distinguish. The authors should consider broadening lines and curves to help the readers.

Figure 1C and Figure 6E: If $n=3$, why is there more than 3 points in the graphs? Statistics should be conducted on biological replicates, not technical ones.

Figure 2: Panels J and K are confusing. Since the endpoint was met upon reaching maximum tumor volume, the volume is not a relevant measure to show. Panel K: Is a point missing for Control cells? Prism usually display points this way when 3 points are at a similar level, but only two are visible. Panel L: Staining of p-H3 in 22RV1 cells seems different in RT and CANA+RT samples. Moreover, the picture looks stretched on the x-axis for RT treated tumors using the same staining.

Figure 4: Panel A: As stated before, correction for cell content is crucial to interpret these assays. No normalization is stated in the title of y axis, and no details regarding normalization are given within the MM section. Panel H: The authors should either make gaps clearly visible or re-export the figure to make the panel an integrated unit. Panel K: Given the changes observed in energy metabolism and changes in GAPDH expression (as seen in panel H), GAPDH does not appear to be a reliable internal control for WB loading.

Figure 5: Panel A: Since p-mTOR and pRaptor are shown, mTOR and RAPTOR blots should be shown as well. Panel B: For consistency, the authors should also show data for 2Gy as well.

Figure 6 legend. Line 1007: Please cite letters in order of appearance.

Reviewer #2 (Remarks to the Author):

COMMSBIO-22-3785-T

The SGLT2 inhibitor canagliflozin suppresses growth and enhances prostate cancer response to radiotherapy: Transcriptional reprogramming, suppression of mTORC1- HIF-1 α pathway and radio-resistance at clinically achievable doses.

by Ali et al.

To treat prostate cancer patients, radiotherapy is one of the standard of care therapies despite resistance to the treatment and its impact on the quality of life of these patients. Therefore, there is an urgent clinical need to improve the protocols including combination of radiotherapy with agents that could increase its efficacy. In this context, the authors aim to investigate the potential benefit of combining radiotherapy with the use of canagliflozin, an inhibitor of sodium-glucose co-transporter-2. This drug has been already approved in the case of diabetes and heart failure and shown to reduce prostate cancer growth.

The relevance of such a project is unquestionable. Furthermore, drug repositioning is an interesting strategy to accelerate the entry of new therapeutic strategies into the clinic. However, the manuscript is a collection of many data that do not follow a clear thread. Reported results arise from too many different experimental conditions. Different models (starting from a panel of 5 cell lines for the first in vitro studies to focus on two for the transcriptomic and in vivo analysis) appear without a clear rationale. Drug and radiotherapy doses as well as the time of exposure change in an apparently random and unargued way. The manuscript is confusing, and it is not sufficiently argued (are the signaling pathways reported to be altered the drivers or the consequences of drug and/or radiotherapy effects?). Finally, correlations are used to draw causal conclusions and thus, such conclusions are overstated.

Reviewer #3 (Remarks to the Author):

The paper presents a new insight in suppressing prostate cancer (PrCa) growth and enhancing its response to radiotherapy (RT). It is a topic of interest for PrCa standard oncology treatment area because PrCa tends to develop a resistance to RT in late stages, herein, the article finds out that a sodium-glucose co-transporter-2 (SGLT2) inhibitor, canagliflozin, manages to synergize with or sensitize PrCa to RT. The paper examine canagliflozin's efficacy, analyze the mechanism of action of drug, and carried out the results that harnessing canagliflozin by suppression of the mTORC1-HIF-1 α pathway, DNA replication and the cell cycle reduces PrCa progression, improves RT sensitivity, and causes a marked reprogramming of PrCa transcriptional activity. Overall, the study is accessible, logical, and well supported by strong data sets. It will be suitable for publication if the minor concerns below can be addressed. It is mentioned in line62 that RT dose-escalation is related with bowel and genitourinary complications. Therefore, the article would be more complete if the data on the improvement of the patient's intestinal symptoms were included. In addition, there are many clinical trials in recent years in the field of immunotherapy of prostate tumors combining immune checkpoint inhibitors with radiotherapy, and it would be enlightening if these trials are mentioned in the conclusion part.

Response to Reviewer Comments

The SGLT2 inhibitor canagliflozin suppresses growth and enhances prostate cancer response to radiotherapy. [Transcriptional reprogramming, suppression of mTORC1-HIF-1 α pathway and radio-resistance at clinically achievable doses]” (Manuscript Number: COMMSBIO-22-3785-T).

Dear Reviewers

We greatly appreciate your time to review our manuscript and thank you for all your thoughtful comments. They were very helpful in revising our manuscript. Below we present a point-by-point response to your comments. Changes can be reviewed in the tracked version of the manuscript as well as the non-tracked version in red color.

Reviewer #1

(Remarks to the Author) (Expertise: cancer cell metabolism, canagliflozin): The paper by Ali and colleagues uncovers a role for the anti-diabetic drug canagliflozin in promoting sensitivity to radiotherapy in prostate cancer. The paper presents data that will stimulate studies on anti-cancer combination therapies incorporating canagliflozin. While the paper is well written in general, the manuscript could be improved by presenting a more integrated and cohesive story.

Also, the manuscript would be improved by discussing the results and incorporating more recent literature on prostate cancer and canagliflozin. Please find below specific comments.

Response: We have revised the introduction to include more recent references and ensure that it accurately reflects the current state of research in the field. Details of changes made are described below.

The mechanism of action presented for canagliflozin is that it inhibits complex I. Yet, combining canagliflozin with two other complex I inhibitors augments its anti-proliferative properties. It would be important to explore whether canagliflozin could inhibit other complexes in the electron transport chain. For example, one could examine whether combining canagliflozin with complex II inhibitors (malonate) or complex III inhibitors (myxothiazol/Antimycin A) augment its anti-proliferative properties. It would also be interesting to examine whether the anti-proliferative effects of canagliflozin are augmented by treatment with the anti-diabetic drug metformin that also inhibits complex I.

Previous studies from members of our group and others (PMID: 27381369, and PMID: 29445145) showed that canagliflozin is a specific ETC complex-I inhibitor. We were the first group to demonstrate the anti-cancer effects of canagliflozin (PMID: 27689018). We showed that canagliflozin inhibits complex-I-supported respiration in PC3 cells through inhibition of mitochondrial ND1. We also investigated in the same study whether canagliflozin affects complex-II using permeabilized PC3 cells and discovered that Canagliflozin dose-dependently reduced oxygen consumption via complex-I inhibition without affecting complex-II. In the same study we

measured complex-II supported respiration in PC3 cells by treating the cells with succinate (10mM), which was then inhibited by the complex-II specific inhibitor **malonate** (5mM). Then, the difference in respiration between succinate-stimulated and residual respiration after malonate treatment was then analyzed (PMID: 27689018).

Secker et al (PMID: 29445145) compared the effect of canagliflozin on complex-I, -II, and -IV activity in mitochondria by exposing cells to canagliflozin or the complex-II inhibitor (thenoyltrifluoroacetone; **TTF**A) or complex-IV inhibitor (**NaN3**). They found that canagliflozin only inhibited mitochondrial complex-I but had no effect on complex-II or -IV -supported respiration. Specifically, Secker et al (PMID: 29445145) confirmed that canagliflozin inhibited ETC complex-I specificity by preparing mitochondrial-enriched fractions of RPTEC/TERT1 cells and quantifying **Rotenone-sensitive NADH oxidation** (equivalent to complex-I activity) as well as **Atpenin A5-sensitive succinate oxidation** (equivalent to complex-II activity). They found that canagliflozin reduced NADH oxidation but not succinate oxidation, confirming that canagliflozin inhibits mitochondrial ETC through complex-I inhibition but not Complex-II.

Therefore, we feel that there is adequate information on this topic to indicate that canagliflozin inhibits complex I alone. A brief statement is now inserted into the text of the manuscript.

(Line 279-280):

“Further, the work of our group and others indicated that canagliflozin inhibits specifically complex I but not complex II-IV.”

We agree with the reviewer that investigation of combined canagliflozin + metformin treatment is an interesting question. We did examine this question earlier, but we found no evidence of significant additivity between the two drugs in terms of their anti-proliferation activity, as illustrated in the figure below.

It is strange that ECAR does not increase upon OCR inhibition. This is a central node of bioenergetics regulation in cells. In order to assess the impact of canagliflozin on cellular energy stress, the levels of ATP, ADP and AMP should be quantified. The authors should also specify the buffering correction that was used for the Seahorse experiments.

A) “It is strange that ECAR does not increase upon OCR inhibition. This is a central node of bioenergetics regulation in cells.”

We agree with the reviewer. This observation was unexpected, but it is an important finding indeed. The results of **Figure 4f** show the difference between canagliflozin and the specific OxPhos inhibitors in terms of their effects on proton efflux rate (PER). Those results are in agreement with those of **Figure s3c** illustrating that canagliflozin, unlike its effects on OxPhos genes (**figure 3d**), did not alter the expression of glycolytic genes and appeared to suppress the expression of monocarboxylate transporters such as SLC16A14 supporting the transport of molecules like lactate across the plasma membrane.

Figure s3c

(Line 292-298):

“The inhibition of OCR observed in response to canagliflozin treatment is likely responsible for the induction of genes encoding subunits of mitochondria respiratory chain enzymes and part of a feedback response (**Figure 3d**). On the other hand, the lack of significant change in ECAR rate is consistent with the lack of upregulation of glycolysis related genes (**Figure s3c**). Interestingly, canagliflozin suppressed the expression of a number of transporter genes including SLC16A14, a gene believed to facilitate transport of H⁺ and monocarboxylates, like lactate, across the plasma membrane⁴⁸ (**Figure s3c**).”

(Line 299-308):

“**Canagliflozin vs OxPhos inhibitors:** To determine whether canagliflozin mediates its action solely through OxPhos inhibition, we compared its anti-proliferative activity to that of two specific OxPhos complex-I inhibitors in clinical development, BAY-87-2243⁴⁹ and IACS-010759⁵⁰. At widely used doses, these agents significantly suppressed OCR in glycolytic rate bio-analyzer assays (**Figure s3e**). However, unlike canagliflozin, BAY-87-2243 and IACS-010759

significantly increased the proton efflux rate (PER) (**Figure 4f**). Despite a more effective suppression of OCR, we observed a limited 30-40% inhibition of PC3 cell proliferation by these agents, compared to 45% and 80% with 10 or 30 μ M canagliflozin, respectively. Further, canagliflozin induced additional anti-proliferative efficacy when combined with BAY-87-2243 and IACS-010759 (**Figure 4g**).”

B) “In order to assess the impact of canagliflozin on cellular energy stress, the levels of ATP, ADP and AMP should be quantified”.

We agree with the reviewer that such assays are important when investigating a new metabolic agent. However, members of our group measured the ADP:ATP ratio treated with canagliflozin in previous studies. Hawley et al. (PMID: 27381369) reported that HEK-293 cells treated with CANA for 1 hour demonstrated a significant increase in cellular ADP:ATP ratio, as determined by capillary electrophoresis of perchloric acid extracts.

Further, Hawley et al. (PMID: 27381369) showed that canagliflozin activates AMPK by increasing the cellular AMP/ADP ratio using a comparison of cells expressing wild type AMPK- γ 2 subunit or the AMP/ADP-insensitive R531G mutant. In subsequent studies, we evaluated ATP production after canagliflozin levels in PC3 cells [Villani et al. (PMID: 27689018)]. Suppression of ATP levels and proliferation were potentiated when cells were incubated with galactose, which increases the reliance on OxPhos, compared to glucose. Therefore, it is clear that canagliflozin mediates a measurable suppression of cellular ATP levels. Therefore, we do not feel that repetition of this work adds significant value to the present manuscript.

C) “The authors should also specify the buffering correction that was used for the Seahorse experiments.”

For buffer correction, the default buffer factor correction (2.40) was applied. For Proton Efflux Rate calculations, the default buffer factor was taken into consideration and was applied before calculations. That section has been added to the method section under the title Seahorse Bioanalyzer Assays. (Line 651-652):

“The default buffer factor (2.4) was taken into consideration and was applied before calculations.”

3- Data are presented to show that canagliflozin inhibits mTOR signalling. The impact of combining mTOR inhibitors with canagliflozin on the proliferative potential of prostate cancer cells should be explored.

In response to this comment, we examined the widely used mTOR inhibitor rapamycin (Sirolimus®) alone and in combination with canagliflozin. At the clinically achievable dose of 10nM, rapamycin alone inhibited proliferation by 40-45%. However, combining canagliflozin (10

or 30 μ M) with rapamycin did not yield an additive effect. These results show that blockade of mTOR by clinically achievable doses of rapamycin alone has limited impact of prostate cancer cells proliferation. Addition of rapamycin to canagliflozin did not provide improved anti-proliferative efficacy indicating that the two drugs elicit luckily similar mechanisms in prostate cancer cells.

4- *The use of part of the RNA seq data throughout the paper, while the data are actually discussed in the later parts of the paper, is confusing.*

We appreciate the reviewer's comment. To avoid any confusion, we have modified the following:

1) Rearrangemant of the RNAseq results: We have rearranged the RNAseq results that was breakout through the figures 4-7 and moved them into figure 3. Now, in figure 3 we are presenting the RNAseq results from non-irradiated PC3 cells, a CRPC model with neuroendocrine features, representing a disease entity that is not typically treated with radiotherapy. (Changes were modified in the text as follows:

Lines 239-270:

" RNAseq analysis revealed that canagliflozin at clinical achievable dose (10 μ M), demonstrated dual effects on gene expression in PC3 cells. It mediated feedback upregulation of genes related to autophagy, organelle acidification V-ATPase (ATP6V-0E1, -1D, -1E1, -1G1, -1H), and genes encoding subunits of mitochondria respiratory chain enzymes [complex-I: (ND1, NDUFA1-FV2), complex-II (SDHB-D) and, complex-III (COX1-8A, UQCRB-RQ)] (**Figure 3d**). In contrast, canagliflozin downregulated genes involved in the MAPK-H3 and mTORC1-p70S6k/4EBP1-related genes, including MAPK-Kinase-Kinase-11 (MAP3K11; an activator of B-Raf, Jun-N Terminal-kinase (JNK), Erk and p38 MAPK), MAPKs p38-alpha (MAPK14), and MAPK-Activated-Protein-kinase-3 (MAPKAPK3: a target of Erk1/2), Akt1, Akt2, ribosomal p90-S6-kinase (RPS6KA1) and PI3k regulatory (PIK3R1, PIK3R3) and catalytic (PI3KC3) subunits (**Figure 3e**). Canagliflozin's effects extended to regulating the expression of early signalling

mediators and effectors of tyrosine kinase receptor pathways upstream of MAPKs and Akt, such as adaptor molecules IRS, SHC3 and GRB2, as well as small GTP-binding proteins involved in signal transduction (RAC2) and transcription factors such as EGR2, REST, ELK1, and E2F (including E2F2, E2F3, and E2F4).

Interestingly, we found that canagliflozin downregulated transcription of genes that enhance HIF-1 α degradation such as Protein-Kinase C-alpha (*PRKCA*)⁴⁴. This was associated with downregulation of genes that are directly regulated by HIF-1 α , such as *VEGF* and *TGFB3*^{15,45} (**Figure 3f**). Moreover, the drug increased expression of *ELOB*, *ELOC*, *RBX1*, and *VBPI* which participate in the formation of the VHL-box E3-ubiquitin ligase that targets HIF-1 α for degradation⁴⁶.

Furthermore, canagliflozin mediated substantial modulation of key genes involved in DNA replication and cell cycle progression. Gene Ontology DEG sets of cell cycle phase transition (GO:0044770) and cell cycle progression (GO:0051726) were suppressed by canagliflozin. These included genes involved in G1-S transition (*CDC25A*, *MCMC2*, *CDK2* and *CDC20*), G2-M checkpoint (*GTSE1*, *CCNF* and *ATF5*), M-phase and overall cell cycle and DNA replication regulators (*PLK1*, *E2F* transcription factors, *CDCA8*, *POLD1*, *TIMELESS*), and sensors of cellular stress that actively regulate cell cycle (*MAPK14* and *GADD45B*) (**Figure 3g**).

These results illustrated substantial complexity and involvement of multiple biological processes in canagliflozin's mechanism of action. They demonstrate that within clinically achievable doses canagliflozin alone is able to suppress key molecular pathways supporting survival and tumor progression.”

Line 296-298:

“Interestingly, canagliflozin suppressed the expression of a number of transporter genes including **SLC16A14**, a gene believed to facilitate transport of H⁺ and monocarboxylates, like lactate, across the plasma membrane⁴⁸ (**Figure s3c**).”

Line 361-363:

“These results are congruent with the upregulation of *DUSP5* by canagliflozin, a nuclear MAPK phosphatase and the observed downregulation of MAPK-, mTORC1- related genes and upstream adaptor molecules (**Figure 3e**).”

Figure 3

RNAseq of PC3 treated with canagliflozin (FDR q-value < 0.05)

d Metabolic gene expression

Vacuolar ATPase/H(+)-Transporting ATPase

Control A
Control B
Control C
CAN A
CAN B
CAN C

Mitochondrial OxPhos genes

Control A
Control B
Control C
CAN A
CAN B
CAN C

e MAPK and mTOR pathways

Adaptor proteins

Control A
Control B
Control C
CAN A
CAN B
CAN C

MAPK signaling (ERK, P38, JNK)

Control A
Control B
Control C
CAN A
CAN B
CAN C

AKT/mTOR signaling

Control A
Control B
Control C
CAN A
CAN B
CAN C

Transcription factors

Control A
Control B
Control C
CAN A
CAN B
CAN C

f HIF1-α

HIF1-α degradation

Control A
Control B
Control C
CAN A
CAN B
CAN C

g Cell cycle

Cell cycle related genes

Control A
Control B
Control C
CAN A
CAN B
CAN C

Figure 3d-g now contains heatmap figures.

2) Improve clarity and rationale for the approach: We have now introduced new wording in those sections to help improve clarity and rationale for the approach. Figure 8 now focuses on the comparison between irradiated cells with (5Gy) that were treated with canagliflozin (10 μ M) vs non treated irradiated cells and filtered with an FDR q-value < 0.05, while the complete set of data on transcript levels in all treatment conditions is now shown in **Figure s4a**. Finally, in Figure 8, we focus on the drug's transcriptional reprogramming in irradiated cells using 22RV1 cells a cell line that was most sensitive to combined treatment of radiation + canagliflozin.

Line425-426:

“The above observations indicated the value of understanding canagliflozin’s efficacy in regulating gene expression in irradiated cells.”

Line 422-426:

“Transcriptional reprogramming in irradiated cells: Canagliflozin induces mitochondria metabolism and p53 pathway and suppresses the cell cycle progression and DNA replication genes. The above observations indicated the value of understanding canagliflozin’s efficacy in regulating gene expression in irradiated cells.”

Line 431:

“ ... in irradiated cells with addition of canagliflozin ...”

Line 470:

“...RNAseq datasets from non-irradiated PC3 and irradiated 22RV1 cells...”

Line 1134-1135:

“Number of significantly ($p < 0.05$) upregulated and downregulated genes by canagliflozin (10 μ M) in irradiated cells”

Line 1152:

“ of **(a-b)** non-irradiated PC3 and **(c-d)** irradiated 22RV1 cells...”

Line 1156-1157:

“...comparison between irradiated 22RV1 cells with (5Gy) that were treated with canagliflozin (10 μ M) vs non treated irradiated cells”

5- Line 66: The authors should consider using more recent citations.

Recent citations on the role of metabolic dysregulation in tumour cell survival and resistance to cytotoxic therapy have been introduced (PMID: 34862480).

Line 69-70:

“Work in recent decades illustrated the vital role of metabolic deregulation in tumor cell survival and resistance to cytotoxic therapy^{7,8}.”

6- A) Line 68-69: The authors should consider reframing this. More recent theories have been laid out.

We appreciate this suggestion. We have now edited the introduction to address this concern (lines 69-76).

“Work in recent decades illustrated the vital role of metabolic deregulation in tumor cell survival and resistance to cytotoxic therapy^{7,8}. In normal prostatic tissue, androgen receptor signaling guides utilization of glucose and amino acids through glycolysis and the tricarboxylic acid (TCA) cycle to generate and release citrate in the lumen of prostatic glands⁹. Unlike Warburg’s model¹⁰, which suggested a diminishing role of tumor cell mitochondria, PC cells demonstrate enhanced mitochondria function and use the TCA cycle to convert substrate supply from glycolysis, amino acid influx and protein catabolism to de novo synthesis of nucleotides, proteins and lipids required for cellular growth¹¹.”

B) Line 82: The link is not clear. The authors should introduce HIF and mTOR links with mitochondrial metabolism.

We have now edited the introduction to address this concern (lines 76-92):

“The phosphatidylinositol 3-kinase (PI3k) – Akt - mammalian target of rapamycin (mTOR) pathway provides vital support for this function through regulation of membrane transporters, glycolytic and lipogenic enzyme gene expression, while it regulates protein synthesis through p70-S6 kinase (p70^{S6k}) and eukaryotic translation initiation factor 4E-binding-protein-1 (4EBP1)¹¹. HIF-1 α an established cellular response to hypoxia, supports cell survival in the hypoxic tumor micro-environment, but also operates during normoxia downstream of mTOR¹² to promote angiogenesis, radio-resistance, and metastasis¹³. mTOR facilitates HIF-1 α expression through 4EBP1 and STAT3¹¹, but also stabilizes HIF1 α through p70^{S6k}-dependent inhibition of the phosphatase PP2A. The latter deactivates the HIF-1 α prolyl hydroxylase 2 (PHD2), an enzyme that mediates HIF1 α hydroxylation, subsequently facilitating E3-ubiquitination and degradation¹⁴. HIF-1 α is overexpressed in more than 70% of human cancers and is associated with poor prognosis¹⁵. Conversely, HIF-1 α loss inhibits tumor growth in xenograft studies¹⁶.”

“On this basis, targeting mitochondrial metabolism is an attractive strategy to curtail PC growth. This strategy, however, has additional merit. Mitochondrial inhibition leads to the activation of the metabolic stress sensor AMP-activated kinase (AMPK), a hetero-trimeric enzyme with alpha-catalytic and beta- and gamma-regulatory subunits²⁰.”

7- Line 89 and throughout: The authors should normalize phosphorylation annotation throughout the paper.

We have standardized the annotation of phosphorylation throughout the paper as following:

- a) Line 94 “phosphorylation of P-Raptor(Ser⁷⁹²),”
- b) Line 187-188 “P-H3(Ser¹⁰)”
- c) Line 194 “phosphorylated cleaved-caspase-3 (P-CC3(Asp¹⁷⁵))”
- d) Line 210 “P-CC3(Asp¹⁷⁵)”
- e) Line 312 “P-AMPK- α (Thr¹⁷²)”
- f) Line 313 “increased P-ACC(Ser⁷⁹)”
- g) Line 313-314 “Total ACC and P-ACC(Ser⁷⁹)”
- h) Line 316 “P-ACC(Ser⁷⁹)”
- i) Line 345 “P-Raptor(Ser⁷⁹²)”
- j) Line 346 “P-Akt(Ser⁴⁷³)”
- k) Line 348 “P-Akt(Thr³⁰⁸), P-mTOR(Ser²⁴⁴⁸)”
- l) Line 353 “P-H3(Ser¹⁰)”
- m) Line 358 “P-H3(Ser¹⁰)”
- n) Line 359 “P-Erk1/2(Thr²⁰²/Tyr²⁰⁴)”
- o) Line 732-733 “anti-P-H3(Ser¹⁰), anti-P-ACC(Ser⁷⁹), anti-P-CC3(Asp¹⁷⁵)”
- p) Line 738 “P-H3(Ser¹⁰)”

8- Line 117: *The authors should consider calculating the actual IC50 using a non-linear regression model.*

We calculated the drug IC50 and RT half inhibitory doses using a non-linear regression model and we presented these values in a table in **Figure 1a-b**, showing the IC50 values and half inhibitory radiation doses for each cell line when treated with the drug or radiation alone. Additionally, supplementary **Figure s1g** provides data on the combined therapies from cell proliferation and Clonogenic assays.

Figure1:

Cell line	Canagliflozin IC50 (μ M)	Radiation Max/2 dose (Gy)
LnCaP	20.44 μ M	2.0035Gy
22RV1	21.2743 μ M	2.0812Gy
PC3	19.9234 μ M	3.8922Gy
DU145	40.85 μ M	5.498Gy
DU145-RR	37.08 μ M	5.9387Gy

Figure s1g:

Cell Clonogenic-Canaliflozin IC50					Cell Proliferation-Canaliflozin IC50				
Radiation (Gy)	PC3	22RV1	LnCap	DU145	Radiation (Gy)	PC3	22RV1	LnCap	DU145
0	15.01µM	16.6264µM	16.90µM	20.81µM	0	19.9234µM	21.2743µM	20.44µM	40.85µM
2	15.40µM	11.8672µM	14.90µM	24.7799µM	2	20.8702µM	20.7639µM	20.67µM	42.5µM
4	16.95µM	11.232µM	15.20µM	21.03µM	4	21.93µM	21.98µM	22.2296µM	47.13µM
Canagliflozin (µM)					Canagliflozin (µM)				

9- Line 236: Are OXPHOS measures in Fig4a corrected for cell content?

We apologize for the lack of clarity in this figure. First, we felt that Figure 4 was indeed overcrowded, and that presentation of the original graphs generated by the Seahorse bioanalyzer did not add significant value to this figure, particularly since those data were not corrected for cellular content. Those graphs have now been moved to supplemental data and are labelled appropriately. The remaining of the graph in figure 4 (Current figure 4a-e) represent data correct for cellular content as discussed later below.

Line 1093-1094:

“The data were normalized to cell content, and then expressed as a mean average to 0Gy control.”

Line 1214:

“.... raw data...”.

Line 1219-1221:

“Normalized basal OCR values from the glycolytic rate assay (GRA) experiment in PC3 cells treated with canagliflozin, IACS-010759, or BAY-872243. Cell number normalization was performed using crystal violet for the cells.”

10- Line 242: OCR/ECAR ratio is a measure of metabolic phenotype, not of metabolic stress.

We agree with the reviewer and have modified the statement to: OCR/ECAR ratio, “a marker of metabolic phenotype” (Line 290).

11- Line 529: These “cells” were fixed...

We have corrected it to “These cells were fixed” (Line 579)

12- Line 579-595: Please specify how Seahorse data were normalized.

In methods, under Seahorse Bioanalyzer Assays title, we added the following to describe how we normalized the seahorse data:

Line 646-652:

“To normalize of OCR / ECAR / PER values obtained from each assay to cellular content, after the mitochondrial respiration or glycolytic rate assays, cells were fixed with 10% formalin, stained with 0.5% crystal violet, dried, and measured for absorbance at 762nm using the SpectraMax iD5 system (Molecular Devices, San Jose, California) to determine DNA content. The obtained absorbance values, for each well, were then used to normalize the OCR / ECAR / PER values from the same well. The default buffer factor (2.4) was taken into consideration and was applied before calculations.”

13- Lines 599-601: *It looks like some part of the text was cut out.*

We apologize for this error. We have corrected that in Methods section under **Cell cycle analysis:** (Line 657-658):

“After treatments, cells were incubated for 48 hours until they were 50-60% confluent. Cells were then harvested and washed with cold PBS buffer, fixed in 70% ethanol, and stored at -20°C.”

14- A) Lines 616-618: *Please specify the average amount of reads per sample, and if the analysis was done using single- or pair-end sequencing.*

Apologies for this omission: the following sentence is now inserted in Line 674-675:

“Single-end reads were performed at 1x50bp configuration with the goal of generating an average of 25 million reads per sample.”

B) Furthermore, *raw sequencing data should be made available through public repositories.*

We agree that sharing raw sequencing data is important for reproducibility and transparency. We signed an agreement with *Communications Biology* to make the data publicly available and will submit to GEO upon acceptance.

15- Figures

Overall, the graphs are very small and colors hard to distinguish. The authors should consider broadening lines and curves to help the readers.

We have now modified the figures and increased fold and graph size to improve visibility.

- Firstly, we re-organized the figures and moved some to supplementary content, which has enabled us to increase the size of the remaining graphs and improve the visibility of lines and curves.

- Secondly, we have changed colors of certain figures, such as Figure 1c and Figure 1f, to ensure consistency and improve visibility. We hope these changes address the reviewer concerns.

16- Figure 1C and Figure 6E: If n=3, why is there more than 3 points in the graphs? Statistics should be conducted on biological replicates, not technical ones.

In response the reviewer's comment we have now removed any technical replicates shown in the figures. Figures through the paper now show only biological replicates. For statistical analysis was conducted on only biological replicates.

17- A) Figure 2: Panels J and K are confusing. Since the endpoint was met upon reaching maximum tumor volume, the volume is not a relevant measure to show.

We agree with the reviewer. In response to this suggestion, we have removed the tumor volume and weight data for 22RV1 and PC3 (that was in Figures 2J and 2K and Figures 2H and 2I) from the main figure. These figures have been moved to the supplementary **figures s2a-e**.

B) Panel K: Is a point missing for Control cells? Prism usually display points this way when 3 points are at a similar level, but only two are visible.

Thank you for bringing this issue to our attention. We have identified the problem with Prism software and have resolved it. This issue was possibly during exported the graph, as if the

settings are not properly configured, some points or symbols may be missing or distorted in the exported file.

C) Panel L: Staining of p-H3 in 22RV1 cells seems different in RT and CANA+RT samples.

We have replaced the previous images with more representative ones. However, it is frequently observed that radiation treatment changes nuclear morphology.

D) Moreover, the picture looks stretched on the x-axis for RT treated tumors using the same staining.

Thank you for bringing this issue to our attention. We have now adjusted the images.

18- A) Figure 4: Panel A: As stated before, correction for cell content is crucial to interpret these assays. No normalization is stated in the title of y axis, and no details regarding normalization are given within the MM section.

We have addressed this issue as described above.

B) Panel H: The authors should either make gaps clearly visible or re-export the figure to make the panel an integrated unit.

To address this, we have changed the figure, including adding a gap to separate the groups. We have also moved the glycolysis-related to supplementary **figures s3c**.

C) Panel K: Given the changes observed in energy metabolism and changes in GAPDH expression (as seen in panel H), GAPDH does not appear to be a reliable internal control for WB loading.

On this point we would like to clarify that Figure s3c (previously in Figure 4h, describing RNAseq analysis of glycolysis genes, does not demonstrate a convincing change in GAPDH transcript levels (P adjusted value: P: 0.994). Additionally, modulation in gene expression does not necessary translate to changes in protein levels. Therefore, we do not feel that we have evidence indicating that GAPDH is not a reliable control in these experiments. We include here blots comparing β -actin to GAPDH (from the same experiments in figure 4h), indicating that GAPDH protein levels behave similar to β -actin and are not altered by treatments.

19- Figure 5: Panel A: Since p-mTOR and pRaptor are shown, mTOR and RAPTOR blots should be shown as well. Panel B: For consistency, the authors should also show data for 2Gy as well.

We agree that it would be more consistent to include mTOR total and Raptor total blots. We have included these blots into figure 5a. we also included the 2Gy data for consistency.

20- Figure 6 legend. Line 1007: Please cite letters in order of appearance.

We have cited the letters in the order of their appearance.

Reviewer #2:

Reviewer #2 (Remarks to the Author) (*hypoxia signaling*): COMMSBIO-22-3785-T: *The SGLT2 inhibitor canagliflozin suppresses growth and enhances prostate cancer response to radiotherapy: Transcriptional reprogramming, suppression of mTORC1- HIF-1 α pathway and radio-resistance at clinically achievable doses.* by Ali et al. To treat prostate cancer patients, radiotherapy is one of the standards of care therapies despite resistance to the treatment and its impact on the quality of life of these patients. Therefore, there is an urgent clinical need to improve the protocols including combination of radiotherapy with agents that could increase its efficacy. In this context, the authors aim to investigate the potential benefit of combining radiotherapy with the use of canagliflozin, an inhibitor of sodium-glucose co-transporter-2. This drug has been already approved in the case of diabetes and heart failure and shown to reduce prostate cancer growth. The relevance of such a project is unquestionable. Furthermore, drug repositioning is an interesting strategy to accelerate the entry of new therapeutic strategies into the clinic.

- 1- However, the manuscript is a collection of many data that do not follow a clear thread.
- 2- Reported results arise from too many different experimental conditions. Different models (starting from a panel of 5 cell lines for the first in vitro studies to focus on two for the transcriptomic and in vivo analysis) appear without a clear rationale.
- 3- Drug and radiotherapy doses as well as the time of exposure change in an apparently random and unargued way.

We thank the reviewer for the useful comments on our article. We have now edited the manuscript extensively and improved areas of concern. Below we provide explanations on our experimental approaches and reasons and list steps we took to improve conclusions and the flow of information in the manuscript.

Models: prostate cancer develops as multi-focal and genomically heterogeneous disease and progresses in a similar fashion to accumulate mutations in a number of genes including AR, PTEN and Tp53 and evolve into CRPC that frequently includes early or extensive neuroendocrine features in advanced disease. To enhance the applicability of our work, we investigated here a number of hormone sensitive (CSPC) and resistant (CRPC) lines as well cell models with neuroendocrine differentiation. Since this work focuses on the concept of radio-sensitization we also included in our analysis radio-resistant lines developed by our group. We felt this approach improves the strength of our findings and applicability of our work.
(a paragraph is now inserted into the manuscript discussing these issues (lines: 119-126).

“PC develops as a heterogeneous multi-focal disease in the prostate and progresses in a similar fashion to accumulate mutations in a number of genes including androgen receptor (AR) and the tumor suppressors PTEN and Tp53³⁰. Androgen/castrate sensitive PC (CSPC) disease eventually evolves into castrate resistant PC (CRPC) without early or advanced neuroendocrine features³¹. To enhance the applicability of our work, in this study we analyzed castrate-sensitive (LNCap), androgen-responsive CRPC (22RV1) and neuroendocrine cell lines (PC3 and DU145). We began these studies with the evaluation of the anti-proliferative effects of canagliflozin and radiotherapy (Figure 1a-c).”

For practical purposes we focused our in-vivo work into models representing non neuroendocrine and neuroendocrine prostate cancer. Purposely, in depth analysis of transcriptional regulation by canagliflozin was done with PC3 cells, representing neuroendocrine PC that is typically not treated with radiotherapy and with non-neuroendocrine 22RV1 cells that more closely represent localized PC, which is treated with radiotherapy. We also attempted to pursue xenograft experiments with CSPC LNCaps cells but, unfortunately, it was highly challenging to generate xenografts with this cell line. Other groups have had similar experience (PMID: 19399749, PMID: 32676396, PMID: 35116633).

(a paragraph is now added to line: 215-219).

“To better understand canagliflozin’s anti-tumor mechanism, we performed RNAseq analysis on PC3 cells treated with 10 μ M canagliflozin alone, a concentration within its therapeutic window²⁴. PC3 cells were selected for these studies, a model of metastatic CRPC with neuroendocrine features and aggressive form of PC, which is managed mainly with systemic therapy.”

And (line 422-426)

“Transcriptional reprogramming in irradiated cells: Canagliflozin induces mitochondria metabolism and p53 pathway and suppresses the cell cycle progression and DNA replication genes. The above observations indicated the value of understanding canagliflozin’s efficacy in regulating gene expression in irradiated cells.”

and (line 431)

And “ in irradiated cells with addition of canagliflozin”

Canagliflozin dosing: We tested initially a wider range of clinically achievable canagliflozin doses (0-30 μ M) and for practical reasons focused on 10 and 30 μ M mainly as representative of serum doses achievable through oral vs IV drug delivery.

Paragraph added to line 127-129:

“To ensure that our results reliably reflect canagliflozin’s translational value for cancer therapy, we focused our in vitro work on concentrations of canagliflozin shown to be safely achievable in human serum^{24,25}.”

Radiotherapy dose: The radiotherapy doses used in this study were not randomly selected. The majority of the study focused on RT doses of 2Gy or 4-6Gy. This is because 2Gy represents conventional fractionation RT and 4-6 Gy represent the dose range of hypo-fractionated and Ultra-hypo-fractionated RT that is also used widely now. Treatment with 5Gy was specifically used in xenograft studies based on previous work from our group which demonstrated that this radiotherapy treatment mediates a 50% inhibition in tumor growth, an effect felt to be optimal for combination studies.

Paragraph is now added to (line 132-134):

“Radiotherapy response was investigated in variety of doses (2-8Gy) but focused mostly on 2Gy representing conventional or 4-6Gy representing hypo- and ultra-hypo-fractionated radiotherapy used clinically in PC³³.”

4- The manuscript is confusing, and it is not sufficiently argued (are the signaling pathways reported to be altered the drivers or the consequences of drug and/or radiotherapy effects?).

Our group investigated canagliflozin’s anti-cancer efficacy extensively. The work we present here confirms that canagliflozin has indeed a complex and multi-target mechanism of action.

We present new evidence that a significant portion of the drug’s activity relates to modulation of HIF-1 α levels (Figures 7), an established mediator of cellular survival and resistance to radiotherapy. Additionally, we present here evidence of substantial regulation of the key growth and survival promoting pathways of MAPK and mTOR, the latter of which includes HIF-1 α as a downstream target. Indeed, it is challenging to dissect the cause – effect relationship in the mechanism of action of a drug with as wide range of anti-tumor effects as canagliflozin.

The main purpose of this study was to demonstrate whether canagliflozin has adequately potent and mechanistically promising anti-tumor activity to warrant clinical investigation of this agent in prostate cancer in combination with radiotherapy. For that, we felt that it would be a valuable contribution of this study to help understand the breadth of tumor suppressive actions of this agent.

Here, we provide evidence for a significant role of the HIF-1 α pathway. Future studies could focus on dissecting further additional signaling pathways.

5- Finally, correlations are used to draw causal conclusions and thus, such conclusions are overstated.

We have taken steps to correct the language used throughout the manuscript to avoid overstatements.

The modification of statements in lines as follows:

Abstract: line 44-46: “Canagliflozin mediates transcriptional reprogramming over several metabolic and survival pathways known to be regulated by ETS and E2F family transcription factors”

Introductions: lines 110-114: statement was modified to avoid overstatement:

“We show that within its therapeutic window canagliflozin suppresses growth and enhance radiotherapy response in human androgen-sensitive and insensitive PC cells, and tumors, through suppression of the mTORC1-HIF-1 α pathway. Canagliflozin modulates favorably multiple growth and survival pathways and mediates a marked reprogramming of PC transcriptional activity.”

Result and discussion: line 215-219:

“To better understand canagliflozin’s anti-tumor mechanism, we performed RNAseq analysis on PC3 cells treated with 10μM canagliflozin alone, a concentration within its therapeutic window²⁴. PC3 cells were selected for these studies, a model of metastatic CRPC with neuroendocrine features and aggressive form of PC, which is managed mainly with systemic therapy alone.”

Result and discussion: line 419-420:

“Based on these findings we conclude that cell cycle regulation is unlikely to be a key pathway in mediating drug action when canagliflozin is combined with radiation.”

Result and discussion: line 467-468:

“Canagliflozin’s transcriptional regulation points to involvement of ETS and E2F family transcription factors. Association with PC prognosis”

Result and discussion: line 510-512:

“Overall, our data points to ETS and E2F family transcription factors as potential mediators of canagliflozin’s transcriptional program, which regulates favorably genes associated with PC prognosis.”

Figure 10 legend line 1166-1182:

“Figure 10. Cellular pathways modulated by Canagliflozin in prostate cancer cells. The work presented here suggests that canagliflozin mediates its anti-tumor activity through a complex multi-target pathway. Canagliflozin inhibits mitochondria respiration through complex-I, leading to activation of the metabolic stress sensor AMPK. This leads to suppression of ACC-de novo lipogenesis pathway (by increasing the P-ACC(ser79), the inactive form of ACC), Akt/mTOR/p70S6K – 4EBP1 – protein synthesis axis and reduction of HIF-1α levels. Complex I inhibition and suppression of HIF-1α levels contribute to at least part of canagliflozin’s anti-tumor activity. Canagliflozin upregulates the E3-ubiquitine p-VHL complex genes, which may contribute to enhanced HIF-1α degradation. On the other hand, canagliflozin deactivates Erk signaling, likely in part through induction of the nuclear and dual-phosphatase gene (DUSP5). Consistently, the drug blocks, phosphorylation of Histone3 p-H3(ser10), which controls DNA replication and induction of cell cycle checkpoints. Canagliflozin mediates a marked re-programming of prostate cancer transcriptional activity likely through regulation of ELF3, E2F, and ELK transcription factors known to regulate OxPhos, phosphatase, apoptosis, fatty acid metabolism, hypoxia and Wnt pathway genes.

Overall, canagliflozin suppresses biosynthetic, survival and radio-resistance pathways at the transcriptional and post-translational level and mediates tumor suppression and improvement of response to radiotherapy.”

Figure 10: We re-draw figure 10.

Reviewer #3:

Reviewer #3 (Remarks to the Author) (targeting signaling pathways in prostate cancer):

1- The paper presents a new insight in suppressing prostate cancer (PrCa) growth and enhancing its response to radiotherapy (RT). It is a topic of interest for PrCa standard oncology treatment area because PrCa tends to develop a resistance to RT in late stages, herein, the article finds out that a sodium-glucose co-transporter-2 (SGLT2) inhibitor, canagliflozin, manages to synergize with or sensitize PrCa to RT. The paper examines canagliflozin's efficacy, analyze the mechanism of action of drug, and carried out the results that harnessing canagliflozin by suppression of the mTORC1-HIF-1 α pathway, DNA replication and the cell cycle reduces PrCa progression, improves RT sensitivity, and causes a marked reprogramming of PrCa transcriptional activity. Overall, the study is accessible, logical, and well supported by strong data sets. It will be suitable for publication if the minor concerns below can be addressed.

We thank the reviewer for the positive and insightful review of our manuscript.

2- It is mentioned in line62 that RT dose-escalation is related with bowel and genitourinary complications. Therefore, the article would be more complete if the data on the improvement of the patient's intestinal symptoms were included.

Regarding this comment, we admit that we are not completely certain whether the reviewer felt that 1. We should provide the rates of bowel and genitourinary complications in patients receiving dose escalated radiotherapy or 2. discuss potential benefits canagliflozin treatment could offer in terms of reducing bowel and genitourinary complications.

Given that this is a preclinical study that investigated only the potential of canagliflozin to improve radiation therapy response in prostate cancer models and not in patients, we assume statement 1. is what is requested. Therefore, we inserted a description of the increase in acute and late toxicity with dose escalated RT (as per Zietman et al, JCO 2010), (Line 64-68).

“In the past two decades, clinical trials focused on improving outcomes by increasing the dose of radiotherapy delivered to the prostate⁴. However, radiotherapy dose-escalation is associated with increased (5-20%) short- and long-term bowel and genitourinary complications (RTOG grade 2 toxicity)^{5,6}. Development of agents that can synergize with or sensitize PC to radiotherapy would be highly beneficial.”

3- In addition, there are many clinical trials in recent years in the field of immunotherapy of prostate tumors combining immune checkpoint inhibitors with radiotherapy, and it would be enlightening if these trials are mentioned in the conclusion part.

Response to point 3: We appreciate the reviewer's suggestion to include information about recent clinical trials in the field of immunotherapy of prostate tumors. Indeed, trials continue to investigate immune-checkpoint inhibitors in prostate cancer. However, in recent years, modern anti-androgen therapies were shown to improve survival outcomes in metastatic prostate cancer in combination with androgen-deprivation therapy or chemotherapy. These agents, which are now being investigated in combination with radiotherapy in metastatic and localized PC, have shown greater promise compared to immune check-point inhibitor therapy. Nevertheless, trials with both

classes of agents are ongoing. We have now briefly discussed this in the **conclusion** part (Lines 525-530):

“In recent years, clinical trials investigated the safety and efficacy of novel therapeutics alone and in combination with radiotherapy to control PC progression. Modern anti-androgen therapy trials demonstrated improvements in survival in combination with androgen-deprivation or chemotherapy and have changed practice in metastatic PC^{64,65,66}. Immune checkpoint inhibitor therapies were also examined with limited success^{67,68}. Ongoing studies examine these therapies in combination with radiotherapy⁶⁹⁻⁷¹.”

And lines 534-536:

“Demonstrating clinical benefit in early phase trials could open opportunities for further investigation of canagliflozin in combination with anti-androgen and immune check-point inhibitor therapies.”

We hope the modifications made to the manuscript meet the reviewer’s expectations and the manuscript is now acceptable for publication.

Thank you

Amr Ali, et al.

REVIEWERS' COMMENTS:

Reviewer #1 (Remarks to the Author):

The authors have addressed most of our comments. Two points need further clarification.

- 1) Some studies do not report inhibition of complex I with canagliflozin in cancer cells (eg. PMID: 33784591). This should be discussed in the manuscript.
- 2) The buffer correction for the Seahorse experiments concerned the respiration measurements. This should be included.

Reviewer #2 (Remarks to the Author):

The authors have addressed most of the concerns and therefore, the manuscript is almost ready for publication. However, there is one point that I consider should be modified: the PC3 cell line is a metastatic prostate cancer line but not an endocrine model as the authors mention in the text.